# Neurodegenerative disease mutations in TREM2 reveal a functional surface and distinct loss-of-function mechanisms

Daniel L Kober[1,2], Jennifer M Alexander-Brett[2], Celeste M Karch[3], Carlos Cruchaga[3], Marco Colonna[4], Michael J Holtzman[2,5,6], Thomas J Brett[2]*

[1]Molecular Microbiology and Microbial Pathogenesis Program, Washington University School of Medicine, St. Louis, United States; [2]Division of Pulmonary and Critical Care Medicine, Department of Internal Medicine, Washington University School of Medicine, St. Louis, United States; [3]Department of Psychiatry, Washington University School of Medicine, St. Louis, United States; [4]Department of Pathology and Immunology, Washington University School of Medicine, St. Louis, United States; [5]Department of Cell Biology and Physiology, Washington University School of Medicine, St. Louis, United States; [6]Department of Biochemistry and Molecular Biophysics, Washington University School of Medicine, St. Louis, United States

**Abstract** Genetic variations in the myeloid immune receptor TREM2 are linked to several neurodegenerative diseases. To determine how TREM2 variants contribute to these diseases, we performed structural and functional studies of wild-type and variant proteins. Our 3.1 Å TREM2 crystal structure revealed that mutations found in Nasu-Hakola disease are buried whereas Alzheimer's disease risk variants are found on the surface, suggesting that these mutations have distinct effects on TREM2 function. Biophysical and cellular methods indicate that Nasu-Hakola mutations impact protein stability and decrease folded TREM2 surface expression, whereas Alzheimer's risk variants impact binding to a TREM2 ligand. Additionally, the Alzheimer's risk variants appear to epitope map a functional surface on TREM2 that is unique within the larger TREM family. These findings provide a guide to structural and functional differences among genetic variants of TREM2, indicating that therapies targeting the TREM2 pathway should be tailored to these genetic and functional differences with patient-specific medicine approaches for neurodegenerative disorders.

*For correspondence: tbrett@wustl.edu

**Competing interests:** The authors declare that no competing interests exist.

## Introduction

The precise molecular determinants and mechanisms underlying neurodegenerative diseases remain uncertain, but recent large-scale genetic sequencing projects have uncovered new candidates associated with these diseases. For example, whole genome and whole exome sequencing has identified point mutations in the gene encoding the protein TREM2 (triggering receptor expressed on myeloid cells 2) that correlate with a significantly increased risk of developing Alzheimer's disease (AD) (*Guerreiro et al., 2013b*; *Jonsson et al., 2013*). In particular, the TREM2 R47H variant is associated with a risk that is similar to that associated with *APOE4*, previously the only well-established risk factor for late-onset AD. This mutation correlates with increased cerebrospinal fluid (CSF) tau levels, a well-established risk factor for AD (*Cruchaga et al., 2013*), and subsequent studies have identified this mutation in patients who have frontal temporal dementia (FTD), Parkinson's disease (PD) (*Rayaprolu et al., 2013*), and sporadic amyotrophic lateral sclerosis (ALS) (*Cady et al., 2014*). A

**eLife digest** Alzheimer's disease is a neurodegenerative disease and the most common cause of dementia – characterized by memory loss and difficulties with thinking, problem solving and language – in the elderly. Individuals with rare mutations in the gene that encodes a protein called TREM2 have a substantial risk of developing Alzheimer's disease in their mid-60s. A different set of mutations in the gene for TREM2 can cause a more severe degenerative brain disease known as Nasu-Hakola disease in much younger people.

Proteins are made up of chains of building blocks called amino acids that need to fold into specific three-dimensional shapes to allow the protein to work properly. TREM2 is a signaling protein that is found on the surface of immune cells in the brain. Mutations causing Alzheimer's and Nasu-Hakola disease result in the production of mutant TREM2 proteins that differ from the normal protein by only a single amino acid. It is not clear how different mutations affecting the same protein can give rise to two distinct neurodegenerative diseases.

To address this question, Kober et al. used a range of techniques to study normal and mutant TREM2 proteins. First, a technique called X-ray crystallography – which makes it possible to construct three-dimensional models of proteins – revealed that the mutations responsible for Nasu-Hakola disease are buried deep within the core of the folded TREM2 protein. On the other hand, mutations associated with Alzheimer's disease lie on the surface of the protein.

Further experiments examined how these mutations alter the properties of TREM2, revealing that mutations linked to Nasu-Hakola disease affect the ability of TREM2 to fold correctly and how stable its final shape is. This results in fewer TREM2 proteins being present on the surface of immune cells. In contrast, mutations associated with Alzheimer's disease make it harder for TREM2 to bind to molecules known as glycosaminoglycans. The Alzheimer's mutations affect a specific part of TREM2 that is not found in other closely related proteins.

The findings of Kober et al. suggest that TREM2 binding to glycosaminoglycans is likely to be important in preventing Alzheimer's disease. The next step following on from this work is to find out exactly how these interactions affect immune cells, which may aid the development of new therapies for this disease.

recent study of more than 1,600 late-onset AD (LOAD) and cognitively normal brains revealed that microglia-specific networks, including those containing TREM2, are those most significantly dysregulated in LOAD (*Zhang et al., 2013*). Thus, even in the absence of rare variants in *TREM2* that increase AD risk, *TREM2*-containing pathways play a significant role in disease. Recent studies in AD mouse models that are deficient in *TREM2* confirm that loss of TREM2 function contributes to classic AD pathology and demonstrates a crucial role for TREM2 in central nervous system (CNS) biology (*Jay et al., 2015*; *Wang et al., 2015*; *Ulrich et al., 2014*). *TREM2*-deficient AD mice display fewer activated microglia, and the absence of *TREM2* prevents microglia proliferation and promotes microglia apoptosis, which was correlated with increased accumulation of Aβ plaques (*Wang et al., 2015*; *Jay et al., 2015*). Microglia in *TREM2*-deficient AD mice displayed less activation and did not engulf Aβ plaques. This impacts the density of Aβ plaques and promotes diffuse Aβ structures, which in turn are more neurotoxic, and contributes to the accumulation of classic AD pathology (*Wang et al., 2016*; *Yuan et al., 2016*). These findings highlight a crucial role for *TREM2* in maintaining CNS homeostasis. Therefore,understanding how these risk variants affect TREM2 function and contribute to the pathogenesis of neurodegenerative diseases is vital to the development of therapies targeting these devastating conditions.

TREM2 is an innate immune receptor expressed on dendritic cells (DCs), resident macrophages such as osteoclasts and microglia, infiltrating (*Jay et al., 2015*) and inflammatory (*Wu et al., 2015*) macrophages, and CSF monocytes (*Colonna and Wang, 2016*). It is a type one receptor protein consisting of an extracellular V-type Ig domain, a short stalk, a transmembrane domain that associates with the adaptor protein DAP12 for signaling, and a cytoplasmic tail (*Figure 1a*) (*Colonna, 2003*). TREM2 has historically been shown to play an anti-inflammatory role *in vitro* by antagonizing the production of inflammatory cytokines from bone-marrow-derived macrophages

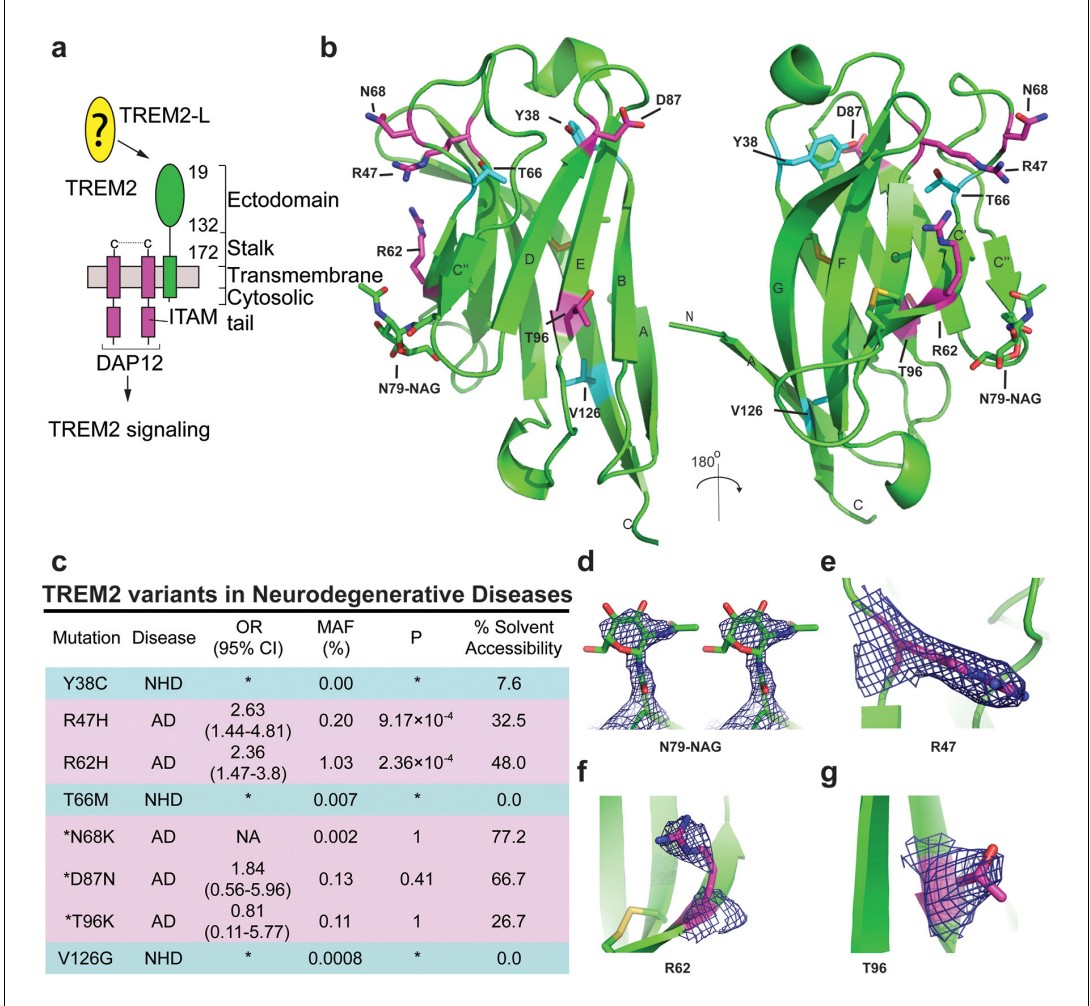

**Figure 1.** Crystal structure of the human TREM2 ectodomain. (a) Schematic of TREM2 cell-surface association with adapter protein DAP12, which contains an Immuno Tyrosine Activation Motif (ITAM). Engagement of TREM2-L by the ectodomain of TREM2 induces signaling. Domain boundaries are indicated. (b) TREM2 ectodomain in two orientations with disease-linked residues shown as sticks. The positions of AD risk variants are shown in magenta, whereas Nasu-Hakola disease (NHD) mutations are shown in cyan. The N-acetylglucosamine (NAG) is shown as green sticks. (c) Table of TREM2 disease-linked mutations, associated disease, and calculated solvent accessible surface exposure for the side-chain (calculated using Naccess), along with statistical correlations to AD (OR = odds ratio; MAF = mean allele frequency) (from *Jin et al., 2014*). Table is highlighted with same color scheme as *Figure 1b*. Validated AD risk variants (R47H and R62H) are not marked. Potential AD risk variants are denoted with an asterisk. (d) Side-by-side stereo view of difference electron density (2mFo-DFc contoured at 2σ) for the N79-NAG. (e–g) Difference electron density (2mFo-DFc contoured at 2σ) for the surface-exposed AD-associated mutation positions (e) R47, (f) R62 and (g) T96.

The following figure supplements are available for figure 1:

**Figure supplement 1.** Analysis of TREM2 glycosylation, comparison of TREM2 monomers in the crystal structure, and SA-omit maps of AD-linked residues.

**Figure supplement 2.** Packing neighbors for the TREM2 AD risk variant R47 and the TREM2 NHD mutants Y38, T66, and V126.

(BMDMs) and dendritic cells (BMDDCs) in response to FcR (*Hamerman et al., 2006*) and Tlr signaling (*Turnbull et al., 2006*; *Ito and Hamerman, 2012*). Likewise, TREM2 participates in phagocytosis of apoptotic cells in cultured microglia and reduces the production of inflammatory cytokines (*Takahashi et al., 2005*). However, TREM2-expressing macrophages can also promote inflammatory disease in the brain (*Jay et al., 2015*) and lung (*Wu et al., 2015*). The identity of a physiologic TREM2 ligand (TREM2-L) remains uncertain, although several classes of molecules have been

proposed, including bacterial carbohydrates (*Daws et al., 2003*; *Quan et al., 2008*), sulfoglycolipids (*Phongsisay et al., 2015*), nucleic acids (*Kawabori et al., 2015*), phospholipids (*Cannon et al., 2012*; *Wang et al., 2015*) and proteins (*Stefano et al., 2009*; *Takegahara et al., 2006*; *Yoon et al., 2012*; *Atagi et al., 2015*; *Bailey et al., 2015*). Additionally, previous studies have identified cells that express a TREM2-L, including astrocytes (*Daws et al., 2003*), DCs (*Ito and Hamerman, 2012*), BMDMs (*Hamerman et al., 2006*), neurons and apoptotic cells (*Hsieh et al., 2009*). This growing body of literature underscores the case for immune deregulation, specifically involving TREM2-associated pathways in neurodegenerative and inflammatory diseases (*Golde et al., 2013*).

Intriguingly, genetic variations in TREM2 are associated with two distinct groups of neurodegenerative diseases. Homozygous mutations including early-stop codons (*Paloneva et al., 2003*; *Soragna et al., 2003*), splice site mutations (*Numasawa et al., 2011*; *Chouery et al., 2008*), the coding stalk mutations D134G and K186N (*Paloneva et al., 2002*), and the coding ectodomain mutations Y38C, T66M, and V126G (*Guerreiro et al., 2013a, 2013c*; *Le Ber et al., 2014*) cause either NHD, characterized by early-onset dementia, demyelination, and bone cyst lipoma (*Paloneva et al., 2002*; *Colonna, 2003*), or a frontotemporal dementia variant with severe loss of brain matter but lacking the bone manifestations. By contrast, *TREM2* mutations associated with AD contribute to disease risk as heterozygous variants. In addition to R47H, the coding mutation R62H is associated with increased risk of AD in independent studies (*Jin et al., 2014*; *Ridge et al., 2016*). These two variants have the strongest risk link to AD. N68K and D87N have also been identified in AD patients, but because these mutations are very rare, their risk remains uncertain (*Guerreiro et al., 2013b*; *Jonsson et al., 2013*). In addition, the mutation T96K has been linked to a decreased risk of AD, but this mutation is also too rare to allow verification of this observation (*Jin et al., 2014*). This curious segregation of disease phenotypes by distinct mutations occurring within the same protein suggest that these mutations should have divergent effects on the structure and function of TREM2.

In order to understand the molecular basis of how different mutations within TREM2 can lead to distinct neurodegenerative diseases, we performed structural, biophysical, and functional studies of wild-type (WT) and mutant TREM2 proteins. To facilitate these studies, we developed a novel mammalian expression system to produce the natively folded and glycosylated TREM2 ectodomains in milligram quantities (*Kober et al., 2014*, *2015*). We determined the crystal structure of the TREM2 ectodomain at 3.1 Å resolution and found that the disease-linked mutations exhibit distinct structural patterns, suggesting that they would impact TREM2 function through alternate mechanisms. To investigate this hypothesis, we carried out extensive biophysical, cellular, and functional assays. We found that NHD-causing mutations impact TREM2 protein folding and stability, whereas AD risk variants decrease binding to the cellular TREM2-ligand (TREM2-L). Furthermore, these AD-linked mutations appear to epitope map a disease-relevant functional surface on TREM2 that facilitates binding to cell-surface TREM2-L. These findings demonstrate two distinct loss-of-function mechanisms for TREM2, illuminate a disease-relevant functional surface on TREM2, and pave the way for the development of patient-specific molecular therapies for the treatment of distinct neurodegenerative diseases.

## Results

### Structure of the hTREM2 ectodomain

We previously reported the crystallization of the WT human TREM2 ectodomain (amino acids 19–134) purified from a mammalian cell expression system (*Kober et al., 2014*). These crystals diffracted to 3.1 Å (*Table 1* and *Table 2*). The TREM2 ectodomain is a V-type Ig domain containing two disulfide bonds, the canonical Ig disulfide between residues C36 and C110 (βB–βF) and an additional linkage between C51 and C60 (βC-βC') (*Figure 1b*). A single N-acetylglucosamine (NAG) glycan that remains after Endo Hf treatment (*Figure 1—figure supplement 1a*) is clearly visible in the electron density at residue N79 (*Figure 1d* and *Figure 1—figure supplement 1g*). The nine β-strands characteristic of proteins within this Ig domain class are present, and two short α-helix passages are observed spanning β-sheets B–C and E–F. The asymmetric unit (ASU) contains two monomers of hTREM2 in a parallel arrangement, but the limited buried surface area (~400 Å²) and chemical nature of the interface suggest that this dimer would not exist in solution (*Figure 1—figure*

**Table 1.** Determination of resolution cut-off by evaluation of meaningful data.

To evaluate the functional resolution of data past 3.3 Å, data were scaled to 3.0 Å resolution. Data were successively truncated at 0.1 Å intervals from 3.4 Å to 3.0 Å and the initial 3.3 Å solution was used to initiate molecular replacement, rigid body refinement and XYZ refinement in PHENIX without manual refinements. After refinement at the higher resolution, the resulting model was used to calculate R and $R_{free}$ values at the immediate lower resolution. If this resulted in a better model, as judged by $R_{free}$, the higher resolution data are useful for improving the model. Data at resolutions that improved the model are highlighted in green while data at resolutions that worsened the model are highlighted in red.

| Calculate at: | | Refine at: | | | |
|---|---|---|---|---|---|
| | | 3.3 | 3.2 | 3.1 | 3.0 |
| | 3.3 | 26.26/31.66 | 25.30/31.47 | | |
| | 3.2 | | 26.75/31.88 | 26.79/31.80 | |
| | 3.1 | | | 27.71/32.47 | 27.64/32.87 |
| | 3.0 | | | | 28.30/32.52 |

supplement 1b). Furthermore, these two monomers are quite similar (Cα RMSD of 0.65 Å, *Figure 1—figure supplement 1c*), therefore, our analysis of hTREM2 described henceforth will refer to chain A.

**Table 2.** Data collection and refinement statistics.

| | Human TREM2 ectodomain |
|---|---|
| **Data collection** | |
| Space group | P $6_4$ 2 2 |
| Cell dimensions | |
| *a, b, c* (Å) | 125.76, 125.76, 183.70 |
| α, β, γ (°) | 90, 90, 120 |
| Resolution (Å) | 50.00–3.10 (3.21–3.10)* |
| $R_{sym}$ | 0.11 (1.00) |
| Mean $I / \sigma I$ | 21.0 (1.76) |
| Completeness (%) | 99.88 (99.75) |
| Redundancy | 12.9 (13.4) |
| **Refinement** | |
| Resolution (Å) | 50.00–3.10 |
| No. reflections | 16,285 |
| $R_{work}$ / $R_{free}$ | 0.2605/0.2736 |
| No. atoms | 1,784 |
| Protein | 1,756 |
| Carbohydrate | 28 |
| *B*-factors | 93.48 |
| Protein | 92.39 |
| Carbohydrate | 162.1 |
| R.m.s. deviations | |
| Bond lengths (Å) | 0.006 |
| Bond angles (°) | 1.18 |

*Values in parentheses are for highest-resolution shell.

We analyzed the structure to map the location of disease-linked point mutations and noticed an intriguing pattern. The side chains of mutations causing NHD (Y38, T66 and V126) are all buried within the core of the Ig fold, whereas the side chains of verified AD risk variants (R47H and R62H) and possible AD risk variants (N68K, D87N and T96K) all lie on the protein surface (*Figure 1b*). Quantification using solvent-accessible surface calculations verified this qualitative observation (*Figure 1c*). Using structural analysis, we can hypothesize how the buried NHD mutations could negatively impact protein folding. Y38 is adjacent to C36, which forms an intramolecular disulfide with C110, so the Y38C mutation most likely disrupts correct disulfide formation (*Figure 1—figure supplement 2b*). T66 immediately follows the C′ β-sheet and is tightly packed inside the core of the protein, with the side chain hydroxyl engaging the backbone amide of K48 (*Figure 1—figure supplement 2c*). The T66M mutation would sterically disrupt this packing and probably destabilize the protein. V126 is located on β-sheet G; it is entirely buried and contributes to a hydrophobic core of the buried residues F24, D104, A105, and Y108. Removal of the side chain by the V126G mutation would disrupt this packing and probably destabilize this hydrophobic core (*Figure 1—figure supplement 2d*). In addition, sequence analysis reveals that Y38 and V126 are conserved within the TREM family, implying that they are probably required to preserve the common fold within this family of receptors (Figure 5c). The T66 residue is less conserved in the TREM family, but is highly conserved in TREM2 proteins across species.

In stark contrast to the NHD mutants, all of the residues implicated in the development of AD are surface exposed (*Figure 1b,c,e–g*) and engage in very few structure-stabilizing contacts. R47, the residue with the strongest link to AD as the R47H variant, is at the end of the loop preceding β-sheet C. It is well-ordered in the structure with the side chain lying parallel along the surface of the protein, and the side-chain amines engage the carbonyl oxygen of T66 (*Figure 1e*, *Figure 1—figure supplement 2d*, and *Figure 1—figure supplement 2a*). R62, another strong AD risk factor as the R62H variant, comes after the loop marked by the C51-C60 disulfide bond (*Figure 1f* and *Figure 1—figure supplement 2e*). Its side chain faces outward and makes no polar contacts. N68 is also surface exposed and its side chain makes no obvious polar contacts within the molecule. D87 is at the end of β-strand D; its side chain extends from the protein surface and the side-chain carboxyl engages the backbone amide of G90. This interaction probably stabilizes the βD-E loop. T96 is on the end of β-strand E and its side chain points away from the surface and engages the side chain of Q33 (*Figure 1g* and *Figure 1—figure supplement 2f*). On the basis of this structural analysis of neurodegenerative disease mutations in TREM2, we hypothesized that NHD mutants (which are buried) would affect protein folding and stability and thus decrease their surface expression, whereas AD risk variants (which lie on the protein surface) would not affect surface expression and instead probably impact ligand binding.

## Impact of TREM2 variants on folding and surface expression

To begin to test tour hypothesis, we expressed each TREM2 ectodomain variant and analyzed their secretion and solution properties using size exclusion chromatography. WT hTREM2 ectodomain consistently eluted as a monomer (*Figure 2a and b*), in accordance with our crystallographic analysis. The verified (R47H and R62H) and possible (N68K, D87N and T96K) AD-risk variants all elute at the same volume as WT TREM2 (*Figure 2—figure supplement 1g–l*), suggesting that these mutations do not drastically alter the folding or oligomerization of TREM2 in solution. Furthermore, all of the TREM2 AD risk variant ectodomains migrated as monomers in the absence of reducing agent (*Figure 2b*). By contrast, although the buried NHD mutants Y38C, T66M, and V126G were secreted, they elute much earlier, consistent with aggregation and misfolding of these proteins (*Figure 2—figure supplement 1m–o*). Consistent with this observation, SDS-PAGE analysis of TREM2 NHD mutants in the absence of reducing agent showed that these ectodomains were largely produced as covalent dimers or trimers, indicating that misfolding of these mutants promoted the formation of aberrant intermolecular disulfide bonds (*Figure 2c*).

Given that our structural analysis suggested that the TREM2 NHD mutants should be misfolded, we were surprised to observe that these ectodomains bypassed cellular quality control and were secreted. We therefore devised an assay using flow cytometry to investigate whether these point mutants in full-length TREM2 impair cell surface expression. DAP12 was co-expressed with FLAG-tagged WT or mutant TREM2 full-length proteins and surface expression was measured either by an anti-FLAG antibody or by anti-TREM2 sera raised against our secreted, folded protein. Whole-cell

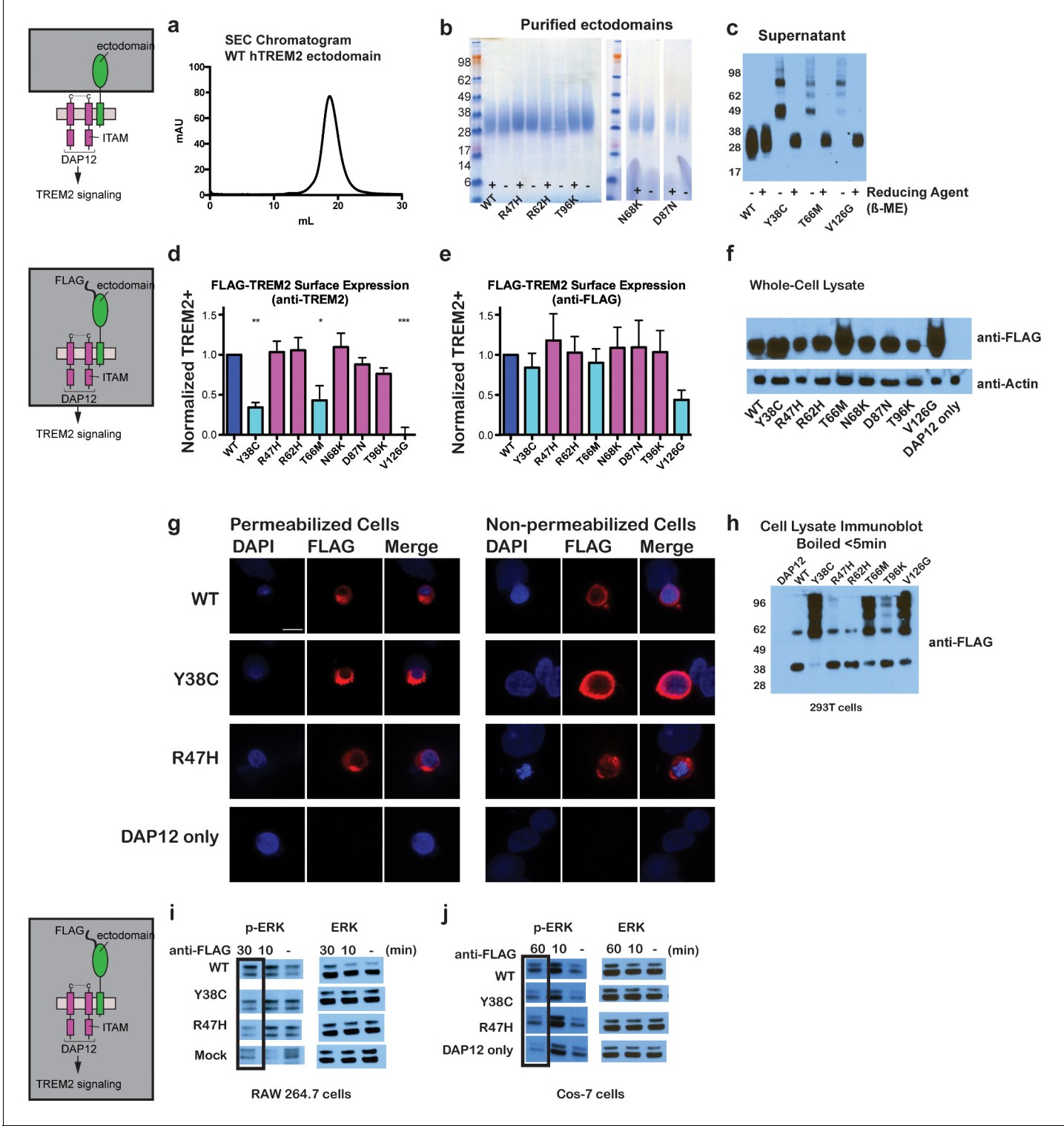

**Figure 2.** Chromatographic and surface expression analysis of WT and variant TREM2. The schematic depicts the domain of TREM2 used in the experiments shown in (a–c). (a) Gel filtration chromatography profile (Superdex 200 10/300 GL) of purified WT human TREM2 ectodomain showing a single monomeric peak. (b) SDS-PAGE analysis of purified TREM2 AD risk variants prepared with (+) and without (-) reducing agent (β-mercaptoethanol). (c) Western blot analysis of TREM2 NHD mutants secreted from transfected 293F cells prepared with (+) and without (-) reducing agent (β-mercaptoethanol). The NHD mutants Y38C, T66M, and V126G all migrate as higher MW oligomers under non-reduced conditions, indicating that they are misfolded and linked by aberrant intermolecular disulfide bonds. Purified WT TREM2 is shown for comparison. Representative of more than three independent expressions. The second schematic depicts FLAG TREM2 used in the experiments shown in (d–h). (d–e) Surface expression of

*Figure 2 continued on next page*

*Figure 2 continued*

WT and variant TREM2 in 293F cells assayed by flow cytometry. Full-length FLAG-WT or mutant TREM2 were co-transfected with mDAP12 into 293F cells. Surface expression was measured by either (**d**) a polyclonal serum or (**e**) an anti-FLAG antibody. Normalized TREM2 indicates the normalized fraction of cells staining positive over DAP12-only background. Data are from four (**d**) or five (**e**) independent experiments. Bars are color-coded in the same way as the residues in *Figure 1b* (cyan = NHD mutant; magenta = AD risk variant). Error bars are SEM. Significance was determined by ANOVA with Bonferroni post-test correction. (*p<0.05, **p<0.01, and ***p<0.001). (**f**) Western blot analysis of whole-cell lysates of 293F cells used in (**d**) and (**e**) showing expression levels of WT, NHD mutant (Y38C, T66M and V126G), and AD risk variant (R47H, R62H, N68K, D87N and T96K) TREM2. Note that the NHD mutants are more highly expressed than the AD risk variants. (**g**) Confocal microscopy of 293F cells co-transfected with DAP12 and FLAG-TREM2. Cells were fixed and either permeabilized (left) or non-permeabilized (right) and stained with anti-FLAG antibody (red). (**h**) FLAG-TREM2 full-length constructs were co-transfected with DAP12 into 293T cells and expression analyzed by anti-FLAG immunoblot. Samples were prepared by suspending cells in reducing SDS loading buffer and boiling for no more than 5 min. We observe SDS-resistant aggregate bands for the NHD variants, which were largely absent in WT and AD risk variants. The T96K variant shows some light aggregation, consistent with the slight shift (~5°C) in denaturation temperature for that variant (*Figure 3*). Schematic depicts FLAG TREM2 signaling assay employed in the experiments shown in (**i–j**). (**i**) TREM2 signaling analyzed by phosphor-ERK1/2 and ERK1/2 immunoblot in RAW264.7 macrophage cells transfected with WT or variant FLAG-TREM2. (RAW264.7 cells express endogenous DAP12.) 24 hr post-transfection, cells were stimulated with anti-FLAG antibody (1:100) for the indicated length of time. ERK/pERK content was assessed by immunoblot. In these cells, we observe pERK at 10 min post-stimulation. However, only WT sustained signaling 30 min post-stimulation. (**j**) TREM2 signaling analyzed in Cos-7 cells as in (**c**). Cos-7 cells were co-transfected with DAP12 and TREM2. In these cells, there is a non-specific antibody response at 10 min. However, as in the RAW264.7 cells, we observe sustained WT signaling at the later time point which is diminished in R47H and Y38C variants.

The following figure supplements are available for figure 2:

**Figure supplement 1.** Analysis of WT and variant TREM2 ectodomains by size exclusion chromatography.

**Figure supplement 2.** hTREM2 surface expression probed using a commercial antibody.

expression levels of the TREM2 NHD mutants were slightly higher than those of WT or AD risk variant TREM2 (*Figure 2f*). However, as predicted by our structural analysis, anti-TREM2 staining was decreased for the NHD mutants Y38C, T66M, and V126G, but not for the AD risk variants R47H, R62H, N68K, D87N, and T96K (*Figure 2d*). Similar observations were made using a commercial anti-TREM2 antibody (R&D, *Figure 2—figure supplement 2*). By contrast, the anti-FLAG antibody showed surface expression for all TREM2 mutants (*Figure 2e*). We next examined cells expressing TREM2 WT, NHD mutant Y38C, and AD risk variant R47H by immuofluorescence confocal microscopy in the presence or absence of permeabilization. We found that the surface expression pattern in non-permeabilized cells was similar for all cell types, whereas in permeabilized cells, the NHD mutant Y38C displayed diffuse intracellular accumulation, in comparison to WT and R47H which displayed a distinct punctate staining pattern (*Figure 2g*). Taken together, these results suggest that NHD mutant proteins do retain some level of surface expression; but the protein is misfolded or aggregated and thus is not recognized by conformation-specific TREM2 antibodies. Consistent with these observations, we found significant amounts of SDS-resistant aggregates by anti-FLAG immunoblotting when expressing the full-length TREM2 NHD mutants in 293T cells (*Figure 2h*). Finally, we evaluated how these different classes of mutations impact TREM2 signaling by crosslinking the different versions of FLAG-TREM2 using anti-FLAG antibody. Consistent with our observation of FLAG-TREM2 on the cell surface for all the variant proteins, we found that while both the Y38C and R47H variants have some signaling potential, as assessed by phosphorylated-ERK blotting in RAW264.7 and Cos-7 cells, neither of these variants sustains signaling as efficiently as WT TREM2 (*Figure 2i,j*). The loss of sustained signaling by the disease variants may suggest that these mutant proteins have aberrant folding kinetics (and thus are replenished more slowly at the surface than WT proteins) or cannot bind co-factors that may be required to sustain signaling. Altogether, these experiments demonstrate that the buried residue changes linked to NHD cause misfolding and aggregation, with variable impact on surface expression, as would be suggested by our structural analysis.

## AD variants slightly impact TREM2 stability and structure

We next sought to evaluate whether the AD-linked surface mutations affect protein structure or stability using sensitive solution techniques. First, we used circular dichroism (CD) spectroscopy to

analyze whether AD-linked variants induced large conformational changes in the TREM2 ectodomain. For these experiments, we chose to analyze the R47H and R62H mutations, as they represent the most significant TREM2 risk factors identified to date, as well as the T96K mutation, which is a sporadically occurring mutation that is not significantly associated with AD. In order to evaluate whether the point mutations induced conformational changes, we collected CD spectra on the proteins, analyzed for large changes, and compared the ratio in minima at 214 nm (due to β sheets) to minima at 233 nm (due to tryptophans), which are characteristic features of the CD spectra for Ig folds (*Sikkink and Ramirez-Alvarado, 2008*). We found that R62H and T96K displayed CD spectra similar to WT proteins while, surprisingly, R47H showed a subtle, yet statistically significant difference in the 214 nm/233 nm ratio (*Figure 3a and b*). Thus the R47H mutation in TREM2 seems to induce a small, but measurable, conformational change, whereas the R62H and T96K mutations do not induce any structural changes detectable by CD.

We next investigated the impact of these mutations on thermal stability using differential scanning fluorimetry (DSF) (*Niesen et al., 2007*). Each surface variant produced slightly lower denaturation temperatures, with R47H and R62H marginally reduced when compared to WT proteins and T96K having the lowest denaturation temperatures (*Figure 3c*). As a control, the same experiments were carried out in the presence of 1 mM DTT, which induced the expected shift to lower denaturation temperatures by disrupting the disulfide bonds. We also used thermal denaturation CD to assess the thermal stability of these proteins. By obtaining full CD scans at 5°C increments, we observed that TREM2 has a denaturation transition at 225 nm, consistent with other Ig domains

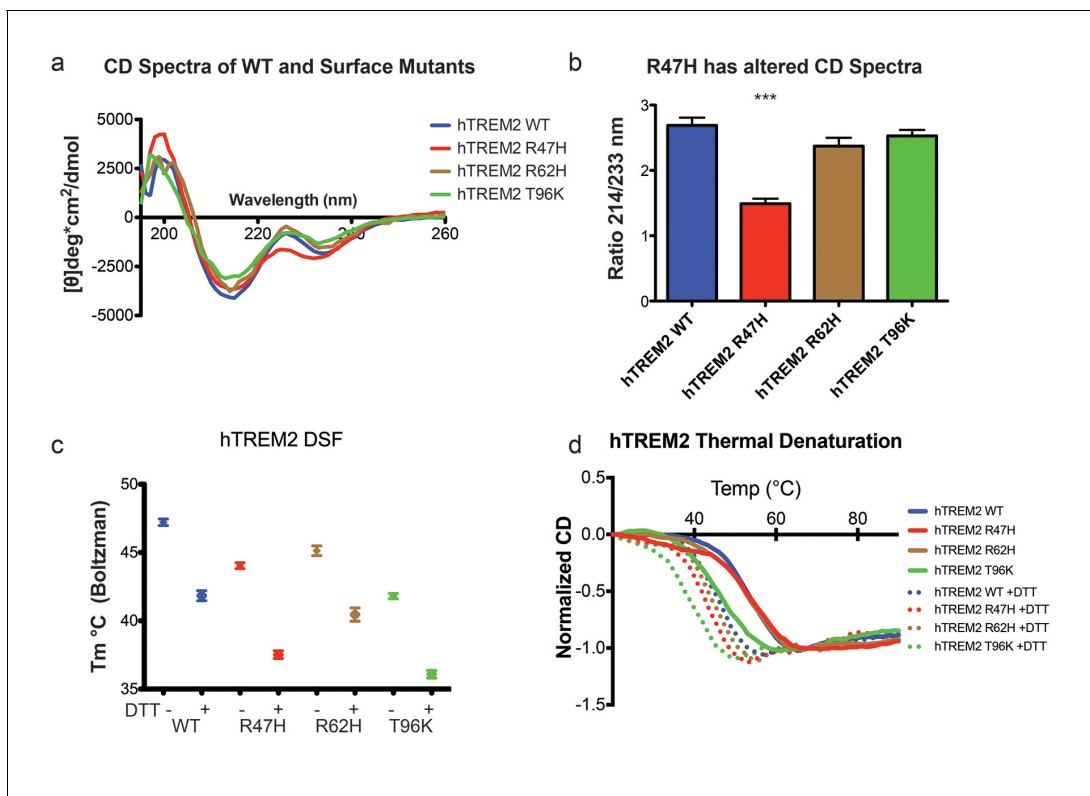

**Figure 3.** Structure and stability analysis by CD and DSF of human WT and AD-linked surface variant TREM2 ectodomains. (**a**) CD spectra of human WT TREM2 (blue) and R47H (red), R62H (brown), and T96K (green) variants. (**b**) Ratio of CD minima at 214 nm and 233 nm reveals that R47H has an altered CD spectrum. Minima ratios were measured on five (WT and R47H) or three (R62H and T96K) independent protein preparations, respectively. (**c**) Thermal melt temperatures measured by DSF. (**d**) WT and variant TREM2 thermal denaturation measured by CD at 225 nm while increasing the temperature from 20°C to 90°C with or without 1 mM DTT.

The following figure supplement is available for figure 3:

**Figure supplement 1.** Structure and stability analysis by CD and DSF of human WT and AD-linked surface variant TREM2 ectodomains.

(*Sikkink and Ramirez-Alvarado, 2008*), and this wavelength was monitored in subsequent denaturation experiments (*Figure 3—figure supplement 1a*). WT TREM2 and all the TREM2 variants tested showed reversible thermal denaturation as they reacquired their native spectra rapidly after cooling from 90°C to 20°C, but their ability to refold was ablated in the presence of DTT (not shown). In contrast to DSF measurements, thermal denaturation by CD did not show a difference between WT, R47H, and R62H proteins; however, the T96K variant has a pronounced shift towards lower denaturation temperature (*Figure 3d*). In the presence of DTT, all proteins had lower denaturation temperatures (*Figure 3d*). We further assayed stability using chemical denaturation by titration of guanidine-HCl (GuHCl) in CD experiments. The R47H and R62H proteins melt at slightly lower GuHCl concentrations than the WT protein, while the T96K variant unfolds at lower GuHCl concentrations than R47H and R62H, consistent with the thermal melt experiments (*Figure 3—figure supplement 1b*). Thus, as predicted by our structural analysis, the AD-risk surface variants do not induce large conformational changes in TREM2, nor do they dramatically impact protein stability, but they can induce subtle changes in both parameters.

## TREM2 AD risk variants impact binding to TREM2-L

Our structural analysis suggested that AD risk variants might impact function by altering ligand binding due to the surface presentation of the mutated residues. Recently, TREM2 was shown to signal following stimulation of reporter cells by plated phospholipids, and the R47H mutation resulted in loss of signaling (*Wang et al., 2015*). To test the direct binding of TREM2 to phospholipids in a cell-free system, we used purified protein in solid-state ELISA and liposome sedimentation assays. Consistent with the results of previous work (*Cannon et al., 2012*), we detected TREM2 binding to phospholipids by both ELISA and liposome sedimentation. However, we did not observe differences in phospholipid discrimination or in direct lipid binding between WT and AD variant TREM2 ectodomains (*Figure 4—figure supplement 1a–c*). We therefore investigated binding to a cell surface ligand (TREM2-L). Although no endogenous protein ligand for TREM2 has yet been identified, several cell types have been reported to express a ligand on the basis of staining with a TREM2-Fc fusion construct (*Ito and Hamerman, 2012*; *Hamerman et al., 2006*; *Daws et al., 2003*; *Hsieh et al., 2009*; *Stefano et al., 2009*). In order to evaluate the effect of AD-linked mutations on TREM2 binding to TREM2-L, we constructed a novel TREM2 protein cell-binding reagent containing a site-specific BirA-biotinylation sequence on the C-terminus of TREM2, which could then be complexed with PE-labeled streptavidin to create a tetrameric cell-staining reagent (*Figure 4a*). In initial experiments, we found that both human and mouse TREM2/SA-PE tetramers were able to bind Neuro2A (N2A) and THP-1 cells (*Figure 4b–d,f,g* and *Figure 4—figure supplement 2a,c*). To validate the specificity of our reagent, we cultured THP-1 cells with various immune stimuli and found that overnight treatment with PMA/ionomycin dramatically decreased binding of TREM2/SA-PE tetramers to THP-1 cells (*Figure 4b and c*), while the myeloid cell marker CD45 remained unchanged (not shown). Other stimuli tested (overnight exposure to IL-13, Poly I:C, LPS, or M-CSF) did not alter staining (*Figure 4—figure supplement 2a*). PMA/ionomycin treatment is the first identified stimulus that ablates TREM2 staining, and this confirms that TREM2 recognizes a specific cell-surface ligand.

We next sought to interrogate the endogenous target recognized by TREM2 on the cell surface, and found that TREM2 binding is sensitive to proteinase treatment of cells prior to incubation with our staining reagent (*Figure 4d* and *Figure 4—figure supplement 2b*). Treatment with proteinase K abolished binding while the more specific proteases chymotrypsin and elastase had an intermediate effect. Proteinase K treatment did not affect cell viability, which was >99% by 7-aminoactinomycin D (7-AAD) staining (*Figure 4—figure supplement 2f*). This demonstrates that TREM2-L contains a protein component. In addition, given the high pI of TREM2, and the frequent reports of anionic potential ligands for TREM2, we reasoned that proteoglycans containing highly sulfated glycosaminoglycans (GAGs) may facilitate cell-surface binding. We tested binding to wild-type CHO-K1 cells, which express heparan and chondroitin sulfates, and to CHO-745 cells, which are deficient in GAG maturation and surface expression (*Esko et al., 1985*). In this experiment, GAG-dependent cell binding of the TREM2 tetramer was observed (*Figure 4e*). Next, we used heparinases (which also cleave heparan sulfates) and chondroitinase ABC to ask whether TREM2 selectively binds either type of GAG. In CHO (*Figure 4e*), THP-1 (*Figure 4f*) and N2A (*Figure 4g*) cells, treatment with heparinases had a pronounced effect on cell surface binding while chondroitinase only slightly

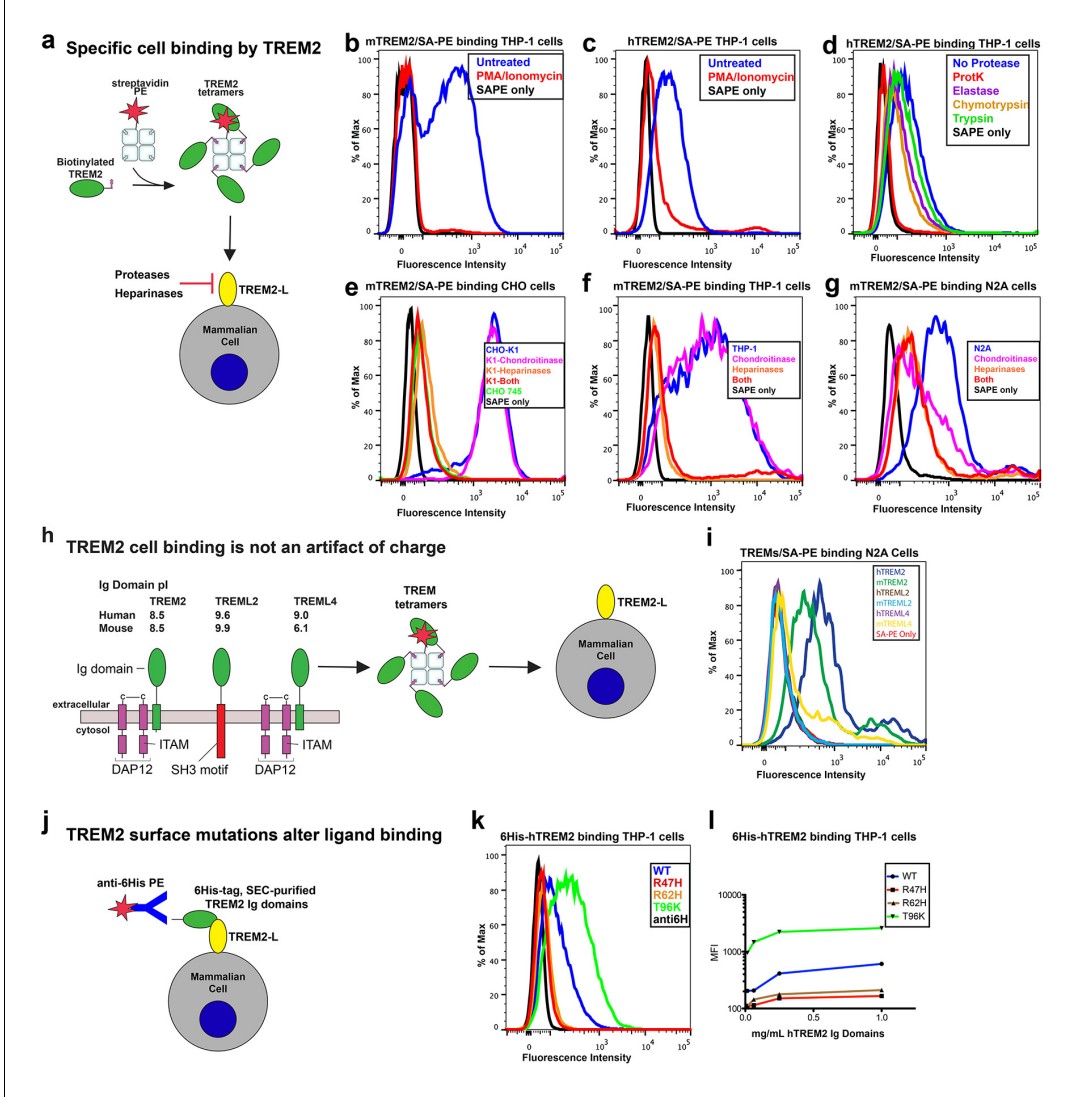

**Figure 4.** AD risk variants in TREM2 alter binding to cell surface TREM2-L. (a) Schematic outlining the flow cytometry experiments in (b–g). TREM2 ectodomains were specifically biotinylated at the C-terminus and probed for binding to TREM2-L on the surface cells, which was reduced by pre-treating cells with proteinases or heparinases. (b–c) Flow cytometry analysis of either (b) mTREM2/SA-PE tetramer or (c) hTREM2/SA-PE tetramer staining of THP-1 cells ± PMA/ionomycin treatment. (d) Flow cytometry analysis of hTREM2/SA-PE tetramer staining of THP-1 cells pretreated by various proteases. (e) mTREM2 staining of CHO-745 (GAGless) and CHO-K1 cells. CHO-K1 cells were treated with a cocktail of heparinases, chondroitinase ABC, or both. (f–g) mTREM2 staining of (f) THP-1 and (g) N2A cells treated with heparinases, chondroitinases, or both. (h) Schematic outline of control cell-binding experiments with the TREM-like proteins TREML2 and TREML4. On the left is a schematic of receptor versions of TREM2, TREML2, and TREML4, along with calculated pI values for the respective ectodomains. (i) Tetramer staining of TREM family ectodomains to N2A cells. (j) Schematic outlining monomeric TREM2-6His cell-staining experiments. Cells were stained with monomeric TREM2-6H and detected using PE-labeled anti-6H antibody. (k) Representative plot of THP-1 staining by hTREM2-6H analyzed by flow cytometry. (l) MFI of anti-6His staining of SEC-purified hTREM2 WT and variant ectodomains pre-incubated with THP-1 cells at the indicated concentrations. All data are representative of at least two independent experiments.

The following figure supplements are available for figure 4:

**Figure supplement 1.** TREM2 lipid binding as assessed by phospholipid ELISA and liposome sedimentation.

**Figure supplement 2.** TREM2 binding to mammalian cells.

diminished binding, together implicating GAGs, specifically heparin sulfate, as a major component of the cell surface TREM2-L.

Next, we tested the impact of TREM2 AD risk variants on TREM2-L binding in three different formats. For these experiments, we chose to compare WT TREM2, the validated TREM2 AD risk variants (R47H and R62H), and the possibly protective AD risk variant (T96K). We utilized anti-6H detection of purified monomeric hTREM2 ectodomains bound to THP-1 cells (*Figure 4j,k, and l*), tetramer staining of WT and variant mTREM2 binding to N2A and THP-1 cells (*Figure 4—figure supplement 2c,d*), and competition assays in which SEC-purified monomers competed with SA-PE-labeled WT tetramers (*Figure 4—figure supplement 2e*). Consistent with our hypothesis, the AD-linked variant R47H displayed markedly diminished binding to N2A and THP-1 cells. In stark contrast, the potentially AD-protective T96K variant significantly increased binding. The R62H variant reduced binding as measured in the anti-6H format, but the reduction in binding was less dramatic when measured by competition or tetramer staining, suggesting an intermediate decrease in affinity. Thus, as suggested by our structural analysis, the AD risk variant R47H TREM2 negatively impacts binding to cell surface TREM2-L, whereas the variant T96K results in increased cellular binding. These data, together with our direct phospholipid binding experiments, suggest that TREM2 AD-risk variants retain phospholipid binding, and instead impact binding to cell surface GAGs. As the R47H risk variant is able to bind phospholipids directly, but does not signal in cellular assays of TREM2 signaling triggered by plated phospholipids (*Wang et al., 2015*), we suggest that TREM2 interaction with GAGs in cis is likely required to orient or cluster TREM2 to mediate signaling upon stimulation by phospholipids. Alternatively, phospholipids may function to orient TREM2 on the cell surface for proper presentation to GAGs and/or additional as of yet undefined protein surface receptors.

## AD risk variants map a unique functional surface on TREM2

Our cell surface TREM2-L binding experiments categorized the naturally occurring TREM2 mutations as loss-of-binding (R47H and R62H) and enhancement-of-binding (T96K) compared to WT, and showed that these mutations influence TREM2 interactions with cell surface GAGs. We returned to analysis of our crystal structure in order to interpret these observations. Upon mapping the electrostatic surface of hTREM2, we noticed a large basic surface that was not present on the surface of other TREM family receptors for which coordinates are available (mTREM1, PDB 1 U9K, [*Kelker et al., 2004a*]; hTREM1, PDB 1SMO, [*Kelker et al., 2004b*]; and TLT-1, PDB 2FRG [*Gattis et al., 2006*]) (*Figure 5a,b and d–f*). Furthermore, sequence analysis shows that the residues constituting this basic patch are highly conserved within TREM2, but not within the rest of the TREM family, suggesting that this interface has evolved specifically toward a role in TREM2 function (*Figure 5c*). To demonstrate this, we prepared staining tetramers for other TREM-like family members with similar overall high pI values calculated for the ectodomain (*Figure 4h*). Consistent with our hypothesis, TREM2 ectodomain tetramers bound to N2A cells, while tetramers of ectodomains from TREM-like 2 (TREML2) and TREM-like 4 (TREML4) did not (*Figure 4i*). This cluster of residues contains most of the AD-linked surface mutations (including R47H and R62H). Additionally, the T96K mutation is located adjacent to this basic patch and would therefore extend it, providing a structural explanation for the gain of function observed with this variant (*Figure 5b*). To test whether additional basic residues located within this surface influence TREM2-L binding, we selected two other basic residues that are conserved only in TREM2 (R76 and R77), and assayed the ability of these variants to bind TREM2-L using our flow cytometry assay. We found that both a R76D and a R77D mutation decreased binding to TREM2-L on THP-1 cells in a manner similar to the validated AD risk variants (R47H and R62H) (*Figure 5g*). Together, the data indicate that this surface represents a conserved functional interface on TREM2 that participates in cell-surface ligand binding and that this surface is unique to TREM2 within the TREM family.

## Discussion

Here we elucidate how distinct point mutations in *TREM2* give rise to different neurodegenerative diseases via two separate loss-of-function mechanisms. Our studies indicate, as presaged by our crystal structure, that NHD-linked mutant residues are buried and impact protein folding and stability, while AD-linked variant residues are on the protein surface and do not diminish the stability or

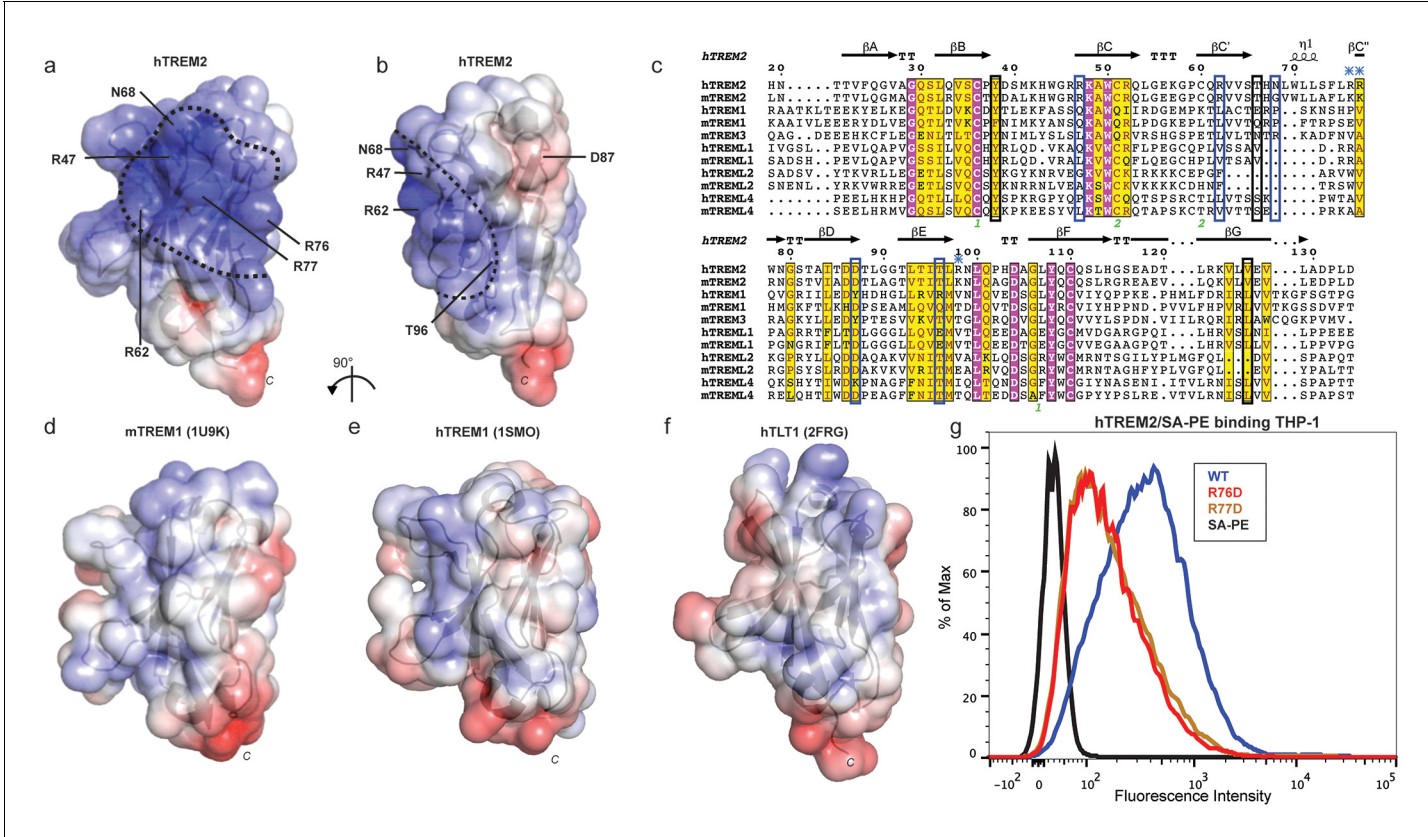

**Figure 5.** AD risk variants reveal a functional surface on TREM2. (**a**) Electrostatic surface of hTREM2 and (**b**) hTREM2 rotated 90°. The extended basic patch on hTREM2 is outlined with a dashed line. (**c**) Structure-based sequence alignment of human (h) and mouse (m) TREM family members. Secondary structure assigned using the DSSP server. Invariant residues are shown in magenta and conserved residues are shown in yellow. Disulfide bonds are numbered in green. NHD residues are highlighted with black boxes and AD residues in blue boxes. Basic patch residues are labeled with blue asterisks. (**d**) mTREM-1 (1 U9K), (**e**) hTREM-1 (1SMO), and (**f**) hTLT-1 (2FRG) aligned with hTREM2 as in (**a**). Electrostatic potential was plotted on the solvent-accessible surface using the AMBER force field and the PDB2PQR server within the APBS Pymol plugin. Scale is −6.0 kT/e (blue) to +6 kT/e (red). (**g**) Flow cytometry analysis of WT, R76D, and R77D hTREM2/SA-PE binding to TREM2-L on THP-1 cells. Representative of two independent experiments.

surface expression of the molecule but instead impact ligand engagement (*Figure 6a*). Several variants have been reported for TREM2, but some are very rare, so it is not clear whether they associate with AD risk. The structure-based studies presented here provide a framework with which one could make predictions as to the variant's impact on function. Our observation of detectable misfolded cell-surface TREM2 NHD mutant proteins was surprising, given that one would expect that the cellular quality-control machinery should degrade misfolded proteins and prevent their surface expression. In addition, previous reports noted decreased surface expression for these mutants (*Park et al., 2015*; *Kleinberger et al., 2014*). Here, we used an approach that probes surface expression using both non-conformational (anti-FLAG) and conformational (anti-hTREM2) antibodies. We found that both the NHD-associated and AD-linked mutant proteins reach the cell surface, however the NHD-associated TREM2 mutants are probably not functional as they are either misfolded or aggregated, and thus are not recognized by the conformational-dependent antibodies. These results are supported by our analysis of the solution behavior of theTREM2 ectodomains of these mutant proteins. An alternative explanation could be that the FLAG tag or overexpression drives the surface expression of these mutated proteins; but we do not think that is the case since we found that the TREM2 ectodomains carrying these mutations (which contain a different tag that is on the opposite terminus) were also secreted. These findings are of potential therapeutic value, as the discovery of

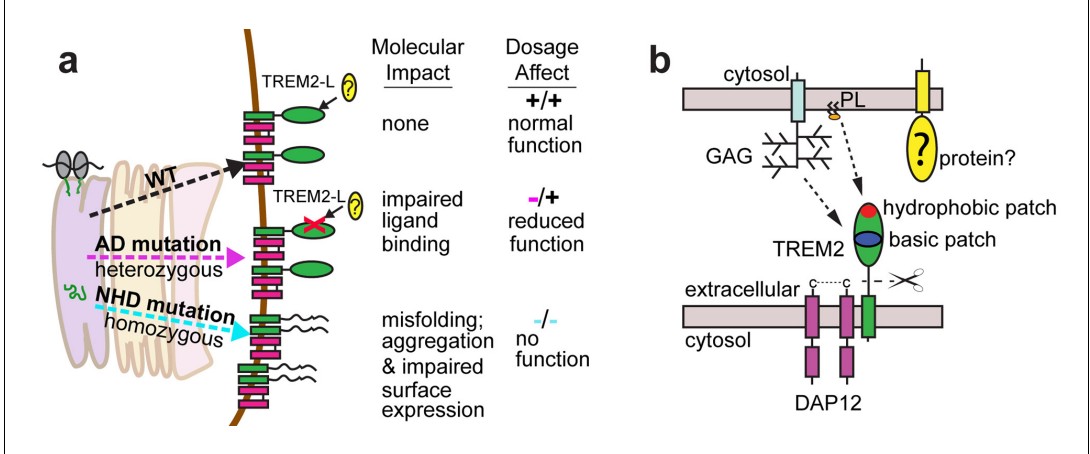

**Figure 6.** Models developed from the current data. (**a**) Current model for the role of TREM2 ectodomain point mutations in neurodegenerative diseases. WT TREM2 is surface expressed in complex with DAP12 and engages TREM2-L normally. AD risk variants do not impair surface expression, but do impair ligand binding. They occur heterozygously, so this would lead to overall reduced TREM2 function in individuals carrying these risk variants. NHD mutations cause misfolding, which leads to aggregation and impaired surface expression. These mutations occur homozygously, leading to no TREM2 function in individuals carrying these mutations. (**b**) Current model for the engagement of possible TREM2 ligands investigated here. Membrane-bound or soluble TREM2 (sTREM2) produced by proteolytic cleavage likely engages GAGs via the basic patch and engages phospholipids (PL) via the hydrophobic patch. In addition, there is a protein component of TREM2-L.

The following figure supplement is available for figure 6:

**Figure supplement 1.** hTREM2 has an extended hydrophobic surface.

molecular chaperones that rescue the folding of and restore function to these mutants could be pursued for patient-specific NHD treatments.

In contrast to the buried NHD mutant proteins, the AD risk variants lie on the surface of the protein and do not severely impact the protein folding or stability of TREM2. However, it should be noted that the R47H mutation does induce a subtle change in secondary structure detectable by CD. The AD mutations probably alter TREM2 function by impairing ligand binding. Recent studies in AD mouse models that are deficient in *TREM2* confirm that loss of TREM2 function contributes to classic AD pathology and suggest a crucial role for TREM2 in microglia function (*Jay et al., 2015*; *Wang et al., 2015*; *Ulrich et al., 2014*). *TREM2*-deficient AD mice display fewer activated microglia, and the absence of *TREM2* prevents microglia proliferation and promotes microglia apoptosis, which is correlated with increased accumulation of Aβ plaques (*Wang et al., 2015*; *Jay et al., 2015*). Microglia in *TREM2*-deficient AD mice display less activation and do not engulf Aβ plaques, which reduces the density of Aβ plaques and promotes diffuse Aβ structures, which, in turn, are more neurotoxic and contribute to the accumulation of classic AD pathology (*Wang et al., 2016*; *Yuan et al., 2016*). These models suggest a crucial role for TREM2 in microglia biology and in the turnover of Aβ. On the basis of the conserved nature of AD-linked basic surface residues and our analysis of cell-surface ligand binding for multiple cell lines, we favor the concept that AD risk variants lead to impaired binding of TREM2-L, which in turn perturbs microglia function.

With this in mind, we investigated how AD risk variants in TREM2 affect binding to previously identified and newly hypothesized ligands based on the analysis of ourcrystal structure. The many diverse ligands reported for TREM2 may in fact suggest that TREM2 is capable of recognizing multiple ligands and that under different conditions, their respective affinities might tune signaling accordingly. For example, low-affinity ligands may induce a tonic inhibitory signal while ligation of a high-affinity ligand could result in an activating response. Indeed, anionic bacterial carbohydrates (*Daws et al., 2003*), phospholipids (*Wang et al., 2015*), myelin lipids (*Poliani et al., 2015*), and even purified DNA (*Kawabori et al., 2015*) all activate TREM2 reporter cell lines. This is not surprising in light of our crystal structure, which reveals an extended basic patch on the surface of TREM2 that would be capable of forming electrostatic interactions with these anionic ligands. In

contrast to highly selective phospholipid binding by other Ig domains (*Kobayashi et al., 2007*; *Miyanishi et al., 2007*; *Santiago et al., 2007*; *Simhadri et al., 2012*), purified TREM2 shows only broad discrimination for anionic (PA, PG, PI, PS) over more neutral (PC, PE) lipids (*Figure 4—figure supplement 1a,b and c*). Structurally, Ig domains with selective lipid binding feature a canonical FG loop motif that contains large hydrophobic residues, absent in TREM2, which mediate membrane insertion and acyl chain binding (*Tietjen et al., 2014*) (*Figure 4—figure supplement 1d,e*). Moreover, while TREM2 does share a conserved aspartic acid residue from this motif, which chelates a required $Ca^{2+}$ ion for the TIM and CD300a proteins, TREM2-lipid binding is not sensitive to EDTA, unlike TIM and CD300 (*Figure 4—figure supplement 1e and f*). Thus, while we do observe direct binding of TREM2 to phospholipids, it does not appear that TREM2 binds phospholipids with high selectivity or using the same binding mode as other phospholipid-binding Ig domains. By contrast, it should be noted that there is a long and narrow hydrophobic patch on the end of TREM2 most distal from the plasma membrane (*Figure 6—figure supplement 1*) that is the most likely lipid-binding surface. Interestingly this patch is adjacent to the basic patch, which could mediate interaction with anionic phospholipid head groups (*Figure 6b*). These hypotheses will need to be tested in future studies using sensitive binding assays (such as surface plasmon resonance) and crystallographic studies of TREM2-phospholipid recognition. In contrast to recent studies employing reporter cells expressing TREM2 R47H (*Wang et al., 2015*), we did not observe any large differences in binding to phospholipids for any of the validated TREM2 AD risk variants. In light of our experimental observations, we believe this is best explained by the hypothesis that another interaction (such as TREM2 binding to GAGs, which we show here is impacted by these AD risk variants) is required to mediate phospholipid signaling; thus in the context of cellular experiments, TREM2 R47H shows diminished signaling, while in direct lipid-binding assays it performs in the same way as WT TREM2. This is potentially supported by a recent report that TREM2 binds to isolated PS, but does not bind apoptotic Jurkat T cells, which present PS on their cell surface but may also be deficient in GAGs (*Bailey et al., 2015*). The results presented here suggest that the point mutations in *TREM2* participate in neurodegenerative disease pathogenesis through distinct molecular mechanisms.

We found that the validated AD risk variants in TREM2 impact binding to a cell-surface ligand (TREM2-L). This ligand does appear to contain a protein component, as pre-treatment of cells with proteases decreases or prevents the binding of TREM2 to these cells. This interaction also appears to be highly GAG-mediated, as TREM2 cell-staining tetramers do not bind CHO 745 cells, which do not express GAGs. This interaction seems most dependent on heparan sulfate, as treatment with heparin sulfatases remove much of the observed binding. As the AD risk variants most notably impact binding to TREM2-L, we reason that this interaction is involved in the pathogenesis of AD. In support of this, we find that the mutations R47H and perhaps R62H deleteriously impact TREM2-L recognition. By contrast, T96K appears to be a potential gain-of-function mutation in our binding experiments. Interestingly, this mutation appears at a much higher frequency than R47H and R62H, and in some studies it has been reported to be enriched in European-American AD cases compared to controls; overall it is associated with a decreased odds ratio for the development of late-onset AD (*Figure 1c*), again suggesting the TREM2-L binding function is coupled to AD risk. Altogether, the structural and functional analysis of these variants illuminates a surface epitope that is specific to TREM2, that is not conserved among the other members of the TREM family, and that mediates binding to a cell-surface ligand. Additional point mutations localized to that surface also resulted in loss of binding, supporting this functional role. Our results suggest that this ligand may have a specific protein component in addition to GAGs. On that note, while this manuscript was in preparation, it was reported that TREM2 binds lipid-loaded ApoE and that this interaction facilitates uptake of Aβ loaded into lipoparticles by microglia (*Yeh et al., 2016*). This report also suggests that TREM2 AD risk variants decrease binding to ApoE, but binding affinities were not quantitated. However, we do not suspect that ApoE is the protein component of cell surface TREM2-L because it is a soluble rather than a surface-bound protein. Our study demonstrates that TREM2 can specifically bind GAGs, utilizing the basic patch; other TREM-like proteins with the same overall calculated pI, but lacking the basic patch residues, do not bind GAGs. We also demonstrate that TREM2 AD risk variants affect binding to GAGs. Future studies will need to focus on how TREM2 binding to GAGs and ApoE are coordinated and how they impact TREM2 function in microglia.

Our results illustrate two distinct paths to loss-of-function in TREM2, so how might these different mechanisms result in either AD or NHD? In attempting to explain this, it is important to take into

account genetics-based dosage affects (*Figure 6a*). In NHD, the mutations are homozygous and the TREM2 is either deleted or misfolded, leading to a complete loss of function, and more serious and early-onset neurodegenerative disease. In AD, the mutations are heterozygous and impair function (but do not completely abolish it), leading to less severe and late onset neurodegenerative disease. Under this hypothesis, it should be possible that a heterozygous NHD mutant would give increased risk for AD. Consistent with this, there have been reports of rare AD cases containing heterozygous NHD variants, including Q33X, Y38C, and T66M (*Guerreiro et al., 2013b*). In addition, TREM2 may function not only as a receptor but also as a signaling molecule as it can be cleaved. We have recently shown that the soluble TREM2 may be a survival signal in bone-marrow macrophages (*Wu et al., 2015*). Interestingly, we have observed that the absence of *TREM2* in an AD mouse model prevents microglia proliferation and promotes microglia apoptosis, suggesting that soluble TREM2 could play a similar role in the CNS (*Wang et al., 2015*). Accordingly, soluble TREM2 can be detected in the CNS, and we have shown that some AD risk variants, but not the NHD mutant proteins are readily detected in the CSF of patients (*Piccio et al., 2016*). Because the AD risk variant proteins are expressed and stable enough to be soluble factors, they may retain some signaling ability and prevent more serious diseases. These hypotheses will require animal knock-in studies for full evaluation. Therefore, a comprehensive analysis of all TREM2 variants will be essential to understanding TREM2 function in both neurodegenerative diseases and other inflammatory diseases, and to design targeted therapies accordingly.

## Materials and methods

### Expression constructs

Full-length wild-type (WT) human *TREM2* as well as R47H, N68K, D87N, and T96K mutants in pMX-3p plasmid and mouse *Trem2* cDNA were used as PCR templates. Mammalian cell expression constructs of human TREM2 wild-type, R47H, D87N, T96K and mouse TREM2 ectodomains were produced as previously described (*Kober et al., 2014, 2015*) by subcloning from these templates into the pHLsec vector, which contains an optimized signal sequence and a C-terminal 6-histidine tag for purification. Similarly, primers were designed to amplify the full-length human *TREM2* gene with an N-terminal FLAG peptide. Additional mutants (human Y38C, R62H, T66M, R76D, R77D; mouse R47H, R62H, and T96K) were generated in the pHsec constructs using either the QuikChange Lightning Site-Direct Mutagenesis Kit (Agilent. Santa Clara, CA) or the Q5 Mutagenesis Kit (NEB) (primer sequences in *Table 3*). WT and mutant human and mouse TREM2 constructs containing site-specific biotinylation sites were generated by *EcoRI-KpnI* restriction digest of the pHLsec inserts followed by ligation into the pHLAvitag3 vector, which encodes a C-terminal BirA biotin ligase biotinylation sequence followed by a 6-histidine tag. All constructs were verified by sequencing.

### Crystallization, structure determination, and analysis

The WT human TREM2 ectodomain was expressed in Freestyle 293F cells in the presence of 1 μg mL$^{-1}$ kifunensine, deglycosylated with EndoHf, and purified for crystallization in a single-step using Ni-NTA resin as previously described (*Kober et al., 2014*). Protein was concentrated to 10 mg ml$^{-1}$ in buffer containing 20 mM HEPES pH 7.4 and 150 mM NaCl. Crystals were grown by hanging drop vapor diffusion by mixing 1:1 with well solution containing 100 mM HEPES 7.0, 2.1 M NaCl, 0.2 M MgCl$_2$ and 0.2 M NDSB-201. Crystals were cryoprotected in mother liquor containing 20% ethylene glycol and flash frozen under a nitrogen stream at −160°C.

Data were collected at the Advanced Photon Source, beamline 19-ID (Argonne National Lab, Chicago). A molecular replacement solution was found with PHASER using mouse TREM-1 ectodomain (1 U9K) (*Kelker et al., 2004a*) as the probe, locating two molecules in the asymmetric unit (ASU). Data were initially scaled and processed at 3.3 Å using HKL2000 (*Otwinowski and Minor, 1997*). The initial model was substantially improved by iterative rounds of manual rebuilding in COOT (*Emsley et al., 2010*) and refinement using Phenix (*Adams et al., 2010*). Next, resolution was extended using the method described by Karplus and Diederichs (*Karplus and Diederichs, 2012*) (*Table 1*). The 3.3 Å model was used as a molecular replacement probe on data scaled to 3.4 Å and 3.3 Å, and subsequently subjected to automated refinement in Phenix without manual intervention. The 3.3 Å solution was then used to calculate Rfree and Rwork at 3.4 Å. Rfree was lower at 3.4 Å

**Table 3.** Primers used in this study.

| | |
|---|---|
| hTREM2 ectodomain | GAAACCGGTCACAACACCACAGTGTTCCAGGGC<br>**CGGGGTACCCAGGGGGTCTGCCAGCACCTCCAC** |
| hTREM2 Y38C* | CTGCAGGTGTCTTGCCCCTGTGACTCCATGAAGCACT<br>AGTGCTTCATGGAGTCACAGGGGCAAGACACTTGCAG |
| hTREM2 T66M* | TGCCAGCGTGTGGTCAGCATGCACAACTTGTGGCTGC<br>GCAGCCACAAGTTGTGCATGCTGACCACACGCTGGCA |
| hTREM2 V126G* | TCAGGAAGGTCCTGGGGGAGGTGCTGGCAGA<br>TCTGCCAGCACCTCCCCCAGGACCTTCCTGA |
| hTREM2 R62H[+] | CCCAGTCCAGCATGTGGTCAGCA<br>CCCTTCTCTCCCAGCTGGC |
| hTREM2 R76D[+] | GTCCTTCCTGGACAGGTGGAATGGG<br>AGCAGCCACAAGTTGTGC |
| hTREM2 R77D[+] | CTTCCTGAGGGACTGGAATGGGAGCACAG<br>GACAGCAGCCACAAGTTG |
| FLAG-hTREM2 | GAAACCGGTGATTATAAAGATGATGATGATAAACACAACACCACAGTGTTCCAGGGC<br>CGGGGTACCTCACGTGTCTCTCAGCCCTGGCAG |
| mTREM2 ectodomain | GAAACCGGTCTCAACACCACGGTGCT<br>CGGGGTACCTTGGTCATCTAGAGGGT |
| mTREM2 R62H[+] | CCCATGCCAGCATGTGGTGAGCA<br>CCCTCCTCACCCAGCTGC |
| mTREM2 R47H* | AAGCACTGGGGGAGACACAAGGCCTGGTGTCGG<br>CCGACACCAGGCCTTGTGTCTCCCCCAGTGCTT |
| mTREM2 T96K* | CTTGCTGGAACCGTCACCATCAAGCTGAAGAACCTCCAAGCCGGT<br>ACCGGCTTGGAGGTTCTTCAGCTTGATGGTGACGGTTCCAGCAAG |
| mDAP12 | CCGGAATTCGCCACCATGGGGGGCTCTGGAGCCCTCCTGG<br>CGGGGTACCTCATCTGTAATATTGCCTCTGTGT |

All primers 5′−3′ with the coding direction primer listed first

\*Designed for Quikchange Lightning Mutagenesis

[+]Designed for NEB Q5 mutagenesis

using the model refined at 3.3 Å compared to the model refined at 3.4 Å, providing evidence that 3.3 Å data improved the model. Similarly, data were again extended to 3.2 Å and finally 3.1 Å. Data past 3.1 Å did not improve the model as determined by an increased Rfree calculated at 3.1 Å using the model refined on 3.0 Å data. Final model refinement occurred by iterative building and refinement in Phenix. NCS and secondary structure restraints were used during refinement, and TLS refinement of B-factors was applied in later rounds. The structure is complete with only one N-terminal residue and four C-terminal residues not visible in the electron density (*Figure 1b*). Ramachandran statistics are 97.7% favored, 0% outliers, 2.3% allowed. Molprobity score was 1.66 (100th percentile) and clashscore was 9.62 (96th percentile) for the final model.

Calculations on the final model were performed using Naccess (*Hubbard and Thornton, 1993*) to measure the solvent accessibility of side chains and HBPLUS (*McDonald and Thornton, 1994*) to identify hydrogen bonds. LigPlot+ (*Laskowski and Swindells, 2011*) was used to analyze side-chain contacts. For structure-based alignment, amino acid sequences were aligned using Clustal Omega (*Sievers et al., 2011*) and residue conservation scored by ESPript (*Robert and Gouet, 2014*). All crystallographic and analysis software used were compiled and distributed by the SBGrid resource (*Morin et al., 2013*) and diffraction images were archived with the SB Data Grid (*Meyer et al., 2016*).

## Cell culture

For protein expression, Freestyle 293F cells were cultured at 8% $CO_2$ in serum-free 293Freestyle media supplemented with Glutamax and penicillin/streptomycin (Pen/Strep, Gibco by ThermoFisher, Waltham, MA). Human monocyte THP-1 cells were cultured at 5% $CO_2$ in RPMI media supplemented with 10% Fetal Bovine Serum (FBS), 10 mM HEPES, 50 μM β-mercaptoethanol, L-glutamine and

penicillin/streptomycin (Pen/Strep). Mouse neuroblast N2A cells were cultured at 5% $CO_2$ in MEM supplemented with 10% FBS, L-glutamine and Pen/Strep. CHO cells were cultured at 5% $CO_2$ in Ham's F12 supplemented with 10% FBS, L-glutamine and Pen/Strep.

## Protein expression and purification

Protein expression for biophysical and functional studies was performed as described previously (*Kober et al., 2014*). In brief, plasmid DNA was complexed at a ratio of 1:2 (μg/μg) with PEI-TMC25 to transfect Freestyle 293F suspension cells cultured in serum-free 293 Freestyle Media. One μg of plasmid DNA was used per $1 \times 10^6$ cells. Transfected cells were allowed to express protein for 72–96 hr. Supernatants were collected and protein was purified using Ni-NTA chromatography. The eluted protein was then further purified by gel filtration chromatography using an analytical s200 size-exclusion column (GE) run in a buffer containing 20 mM Tris pH 8.5 and 150 mM NaCl.

## Surface expression studies

Full-length FLAG-tagged hTREM2 WT or mutant genes in pHL vectors were co-transfected with mDAP12 in pcDNA3.1 vector into 293F cells using 293fectin. After 24 hr, cells were harvested and washed into FACs buffer (1% BSA in PBS). FLAG epitope expression was detected using a FITC-conjugated M2 anti-FLAG antibody (1:50, Sigma-Aldrich. St. Louis, MO), and folded TREM2 surface expression was detected by staining with anti-TREM2 primary polysera (1:1000, hamster) followed by PE-anti-hamster secondary (1:200 eBioscience). Background was defined by cells transfected with mouse DAP12 only.

## WT and mutant TREM2 cell-surface ligand binding studies

### Production of fluorescently labeled TREM2 tetramers

For analysis of cell binding by flow cytometry, pHLAvitag3 constructs of WT and mutant human and mouse TREM2 ectodomains were transfected into Freestyle 293F cells and proteins were purified by Ni-NTA chromatography. The eluted proteins were exchanged into buffer containing 100 mM Tris-HCl pH 7.5, 200 mM K-glutamate, and 5 mM MgCl2, concentrated to 400 μL, and then enzymatic biotinylation was carried out by addition of 40 mM bicine pH 8.3, 8 mM Mg(OAc)$_2$, 8 mM ATP, 0.1 mM biotin and 20 μL *Escherichia coli* BirA biotin ligase (1 mg/mL). Biotinylation proceeded for 12 hr at 4°C. Excess biotin was removed using Zeba desalting spin columns (Thermo Scientific) and biotinylation was confirmed by western blot with Extravidin-HRP (Sigma). The biotinylated TREM2 was pre-incubated with SA-PE at a 4:1 molar ratio for 15 min at 25°C to produce fluorescently labeled tetramers (TREM2/SA-PE). Biotinylated TREM2 was complexed with phycoerythrin-conjugated streptavidin (SA-PE) at a 4:1 molar ratio on ice for 1 hr in 1% FBS prior to staining cells.

### Cell staining

Prior to staining, N2A and CHO cells were lifted following brief exposure to trypsin/EDTA (other cells grow in suspension). Cells ($4 \times 10^5$ cells/ sample) were stained using tetramers at 1:100 dilution in PBS containing 1% FCS (FACS buffer) at 4°C. Cells were either stained with SA-PE alone (1:100) or with TREM2/SA-PE (1:100). Cells were incubated for 1 hr on ice then washed three times with FACS buffer, and then analyzed for binding by flow cytometry using a BD FACScan. Analysis was performed using FlowJo (Tree Star).

### Chemical stimulation

For each condition, $1 \times 10^6$ THP-1 cells were stimulated overnight at the indicated concentration. Poly IC (1 μg mL$^{-1}$), LPS (1 μg mL$^{-1}$), PMA+ionomycin (20 ng μL$^{-1}$ and 500 ng mL$^{-1}$, respectively), M-CSF (20 ng μL$^{-1}$), and IL-13 (50 ng mL$^{-1}$). Cells were collected and washed in FACs buffer before staining.

### Protease treatment

Cells were washed in ice-cold PBS and then incubated with indicated proteases (50 μg mL$^{-1}$) on ice for 30 min. Proteolysis was quenched by washing cells four times in FACs buffer containing PMSF. Cells were stained with SAPE-TREM2 in FACs buffer containing PMSF and HALT protease inhibitors.

Cell viability was verified by 7-AAD staining (BD Biosciences. San Jose, CA ) according to the manufacturer's directions.

## 6-His staining of THP-1 cells

WT and mutant human TREM2 ectodomains were purified by SEC as in *Figure 2—figure supplement 1*. The monomeric peak was collected, concentrated and buffer-exchanged into PBS by centrifugation. Protein concentration was quantitated in triplicate using the Micro BCA kit (Thermo Scientific) according to the manufacturer's instructions. THP-1 cells were washed into FACS buffer and incubated with purified proteins for 1 hr, washed twice, and then incubated in FACS media with Fc-block followed by anti-6His-PE (Miltenyi Biotec. Germany) for 30 min before a final wash and flow cytometry analysis. 20,000 cells were counted for each measurement.

## Proteoglycan digestion assay

Chondroitinase ABC and Heparinases I, II, and III were purchased from Sigma-Aldrich. Digestion of cell-surface sulfated glycans was carried out as follows: cells were washed into PBS with 0.1% BSA and incubated with enzymes at 0.1 U $mL^{-1}$ for 1 hr at 37°C while shaking before being washed three times with cold FACS buffer.

## Thermostability of TREM2 variants by DSF

Thermal stability was assessed by differential scanning fluorimetry (DSF) on protein purified by SEC. The Protein Thermal Shift kit (Applied Biosystems) was used according to the manufacturer's instructions. Briefly, protein was concentrated to 0.5 mg $mL^{-1}$ after buffer exchange into PBS. 5 µl of reaction buffer and 2.5 µl 8x fluorescent dye were added to 12.5 µL protein on ice. Melt-curve experiments were performed using Fast7500 qPCR machine (ABI) starting at 25°C and with continuous 1% ramp to 95°C (roughly 1°C $min^{-1}$). The data were analyzed using Protein Thermal Shift software.

## Thermal and chemical stability of TREM2 variants by CD

Circular dichroism spectroscopy (CD) measurements were performed using a JASCO J-815 spectropolarimeter equipped with a Peltier temperature controller. Thermal denaturation experiments were carried out in 10 mM phosphate (pH 7.0) and 150 mM NaF. For native and GuHCl-denaturation scans, a 1 mm path length cuvette was used and the protein concentration was 30 µM. For denaturation experiments, a 1 cm path length cuvette was used and the protein concentration was 3 µM. For reducing conditions, the buffer contained 1 mM DTT. Ellipticity was measured at 225 nm in 1°C steps from 20°C to 90°C at a rate of 1°C $min^{-1}$ and melt-curve data were smoothed using JASCO software. For chemical stability experiments, purified human ectodomains were incubated with GuHCl (Fluka 50933) at room temperature for at least 1 hr before measuring CD. Repeat measurements at later time points confirmed that equilibrium had been achieved.

## Solid-Phase lipid ELISA

Lipid ELISA experiments were carried out essentially as described by *Kobayashi et al., (2007)*. In brief, phospholipids were dissolved in methanol or methanol:chloroform as needed and diluted to 5 µg $mL^{-1}$ in methanol. 100 µL was added to ELISA plates and allowed to air-dry. Wells were blocked with 3% BSA in PBS. Biotinylated WT or mutant TREM2 in 3% BSA-PBS were incubated overnight at 4°C. Plates were washed three times in PBS + 0.05% Tween 20. Biotinylated TREM2 was detected by streptavidin-HRP (R and D) in 1% BSA-PBS for 2 hr at RT before final wash and developing with TMB Microwell Peroxidase Substrate (KPL. Gaithersburg, MD). Absorbance was measured at 450 nm on a Gemini Plus plate reader (Molecular Devices. Sunnyvale, CA).

## Immunoflourescent confocal microscopy

HEK293T cells were co-transfected with full-length FLAG-TREM2 and DAP12 for 24 hr before replating on glass slides. Cells were washed 2x with PBS, then fixed with 4% paraformaldehyde (PFA) in PBS for 5 min. Cells were then washed and blocked with animal-free blocker (Vector Laboratories. Burlingame, CA) for 1 hr at RT. For permeabilized cells, 0.1% triton x-100 was added to the blocking buffer and subsequent steps. Anti-FLAG antibody (M2, Sigma) was added at 1:1000

overnight at 4°C. Cells were then washed twice, and anti-mouse secondary Alexa Fluor 555 conjugate (Life Technologies) was added at 1:200 for 1 hr. Cells were washed a final time and then mounted in VECTASHIELD H-1200 Mounting Medium with DAPI (Vector Laboratories). Confocal microscopy was carried out using a Zeiss LSM 510 META Confocal Laser Scanning Microscope (Carl Zeiss Microscopy, Thornwood, NY) at 400x magnification. The images were acquired with LSM 4.2 software.

## Liposome sedimentation assays

Liposomes consisted of a base of 35:35:10 PtdCholine, PtdEthanolamine, and cholesterol with an additional 20% wt/wt of candidate phospholipids. Phospholipids were purchased from Sigma or Avanti dissolved in chloroform. Lipids were mixed in solvent-washed glass vials and solvent was evaporated under nitrogen stream. Lipids were resuspended in PBS, warmed at 37°C for 15 min followed by three freeze-thaw cycles on liquid $N_2$ and 37°C $H_2O$. For the sedimentation assay, 10 μg TREM2 proteins quantified by BCA were mixed with 100 μg liposomes and incubated for 1 hr at room temperature. Liposomes were sedimented by centrifugation at 16,800 x $g$ at 4°C. The supernatant was removed and the pellet resuspended in SDS buffer for 6His immunoblot analysis.

## Statistics

All statistics were calculated using GraphPad Prism5.

## Acknowledgements

This work was supported in part by funding from NIH R01-HL119813 (TJB), Knight Alzheimer's Disease Research Center pilot grant P50-AG005681-30.1 (TJB), Alzheimer's Association Research Grant AARG-16-441560 (TJB), K08-HL121168 (JAB), Burroughs-Wellcome Fund Career Award for Medical Scientists (JAB), NIH K01-AG046374 (CMK), R01-AG044546 (CC), R01-AG051485 (MC), R01-HL120153 (MJH), R01-HL121791 (MJH), T32-GM007067 (DLK), and American Heart Association Predoctoral Fellowship PRE22110004 (DLK). Results were derived from work performed at Argonne National Laboratory (ANL) Structural Biology Center. ANL is operated by University of Chicago Argonne, LLC, for the U.S. DOE, Office of Biological and Environmental Research (DE-AC02-06CH11357).

## Additional information

### Funding

| Funder | Grant reference number | Author |
|---|---|---|
| American Heart Association | Predoctoral Fellowship PRE22110004 | Daniel L Kober |
| National Institute of General Medical Sciences | T32-GM007067 | Daniel L Kober |
| National Heart, Lung, and Blood Institute | K08-HL121168 | Jennifer M Alexander-Brett |
| Burroughs Wellcome Fund | Career Award for Medical Scientists | Jennifer M Alexander-Brett |
| National Institute on Aging | K01-AG046374 | Celeste M Karch |
| National Institute on Aging | R01-AG044546 | Carlos Cruchaga |
| National Institute on Aging | R01-AG051485 | Marco Colonna |
| National Heart, Lung, and Blood Institute | R01-HL120153 | Michael J Holtzman |
| National Heart, Lung, and Blood Institute | R01-HL121791 | Michael J Holtzman |
| National Heart, Lung, and Blood Institute | R01-HL119813 | Thomas J Brett |

| National Institute on Aging | P50-AG005681-30.1 | Thomas J Brett |
| Knight Alzheimer's Disease Research Center | Pilot grant P50-AG005681-30.1 | Thomas J Brett |
| Alzheimer's Association | AARG-16-441560 | Thomas J Brett |

The funders had no role in study design, data collection and interpretation, or the decision to submit the work for publication.

## Author contributions

DLK, TJB, Conception and design, Acquisition of data, Analysis and interpretation of data, Drafting or revising the article, Contributed unpublished essential data or reagents; JMA-B, Conception and design, Analysis and interpretation of data, Drafting or revising the article; CMK, Conception and design, Drafting or revising the article; CC, Analysis and interpretation of data, Drafting or revising the article, Contributed unpublished essential data or reagents; MC, MJH, Drafting or revising the article, Contributed unpublished essential data or reagents

## Author ORCIDs

Celeste M Karch, http://orcid.org/0000-0002-6854-5547
Thomas J Brett, http://orcid.org/0000-0002-6871-6676

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
