## [Decision Letter]

Thank you for submitting your article "Neurodegenerative disease mutations in TREM2 reveal a functional surface and distinct loss-of-function mechanisms" for consideration by *eLife*. Your article has been reviewed by three peer reviewers, and the evaluation has been overseen by a Reviewing Editor and Richard Aldrich as the Senior Editor. The following individual involved in review of your submission has agreed to reveal his identity: Todd Golde (Reviewer #1).

The reviewers have discussed the reviews with one another and the Reviewing Editor has drafted this decision to help you prepare a revised submission.

Summary:

This is a biophysical /structural/functional study of the TREM2 ectodomain both from mutant and wild-type TREM2 associated with AD or NH/PLOSL. The bottom line from these studies is the AD associated variants are on the surface of the ectodomain and probably regulate binding of ligands and the PLOSL/NH variants are buried and probably affect folding and maturation as previously discussed. The structural studies appear to be of high quality. However, the other experiments are not optimally or poorly performed, poorly presented, and often over-interpreted, or misinterpreted. The overall recommendation is to re-perform experiments to bring them up to publication quality (e.g., cleanly interpretable gels and analytical SEC traces) or include proper controls, eliminate redundant or irreproducible results, and eliminate unneeded results, as detailed below.

Essential revisions:

1) General/stylistic comment the reference to the Trem2 Ig domain can be confusing. Often the authors refer to an experiment with this domain and then quickly move on to the full-length protein and one has to pay close attention to figure out which was being used. For the reader it would be useful to call the secreted domains something other than TRME 2 Ig which sounds like an immunoglobulin fusion. Maybe just call them soluble TREM2, trem2 ectodmains, or TREM2 (x-Y) x-y being residues of the fusion proteins.

2) Figure 2. There are several concerns:

It would be nice to see the Western blots in the primary data not supplemental. Surface expression should probably be normalized to total protein expression. Immunofluorescence of the cells should show intracellular accumulation of certain mutants that aggregates and this would be a nice addition and simple to do.

The statement "WT hTREM2 Ig domain consistently eluted as a monomer with >95% purity" (based on Figure 2) is not supported by the data. The wt SEC trace shows considerable aggregation, a significant void-volume peak, and shouldering. The PAGE analysis in Figure 2 suggests that this SEC trace is the preparative trace, with contaminating proteins forming aggregates. Please rerun the SEC traces after final purification, and include a complete set of comparable SEC analyses for all the mutants discussed. Also, comparative reduced/non-reduced PAGE is more informative about folding for this intra-chain disulfide-linked protein. Please provide analytical SEC traces for all of the mutants discussed; this is more informative than the PAGE results shown. Why are Westerns shown in Figure 2, and not PAGE gel results? The PAGE results do not necessarily show immature glycosylation – this usually needs comparative EndoH sensitivity analyses to draw that conclusion rigorously.

The results in Figure 2 are inconsistent. Anti-FLAG analyses show consistent levels of cell-surface expression. The authors are making the assumption that the antibodies used are conformational, so the strictly-rigorous conclusion is that these mutant proteins adopt a different conformation, maybe unfolded to some degree, but one that is poorly recognized by these antibodies – cell-surface expression levels appear completely comparable by anti-FLAG. This conclusion also conflicts with the Figure 3 results, which show a "significant" change in the structure of the R47H mutant by CD, which apparently is undetectable by the antibodies used in Figure 2.

3) Figure 3. There are several concerns:

Unless the mutants are confirmed to be monodisperse (free of aggregates, confirmed by analytical SEC analyses, which needs to be shown), then all of the biochemical analyses reported in Figure 3 are not useful. The CD analyses show nice cooperative transitions, so I expect many of these preparations are monodisperse, but this needs to be confirmed. The redundancy in the current presentation by using multiple approaches to assess stability does not improve impact, and are meaningless if the tested protein is aggregating. Overall, there might be subtle differences here but in the current form they do not add a whole lot to the paper.

Figure 3 can be deleted, and the information in Figure 3 can be combined into frames E and F.

Why is the gel shown in Figure 1—figure supplement 2. (a) an immunoblot? Wouldn't a straight PAGE gel be cleaner? The proteins should be available to perform this analysis.

4) Figure 4. There is really no control here. If one takes a protein that is known to bind only to a receptor and does the same study what happens? So, while internal comparisons between the trem2 variants is valid. I think some sort of control for non-specific interactions is needed. Maybe just an Fc domain with that tag or just some other protein ectodomain. Overall, are any of these differences significant? There are no positive and negative binding controls shown, and the results could easily simply represent non-specific binding to sticky lipidic species. Also, what is the point of the rest of this figure, comparing to TIM4? It should be obvious that the PS binding site on TIM4 bears ZERO structural resemblance to the feature being highlighted on TREM2.

5) It is very difficult to make sense of the cell-based binding studies, in that many of the results appear wildly inconsistent. For instance, compare the binding of wt mTREM2/SA-PE to unmodified CHO cells in Figure 4 vs. 4I, or the level of hTREM2/SA-PE binding to untreated THP-1 cells in Figure 4 vs. 5G. These analyses appear too irreproducible to be reliable, suggesting that they may be artifactual.

6) From a genetics perspective, NHL cases are homozygous and AD cases are heterozygous. Indeed, some of the same mutations that cause NHL in their homozygous form lead to AD in the heterozygous form. While the authors do provide convincing evidence that the NHL and AD mutations are located in different parts of the 3D structure of the protein, they seem to ignore the obvious dosage effect. (It is discussed briefly in the last paragraph, but this is far too important an effect to be dismissed lightly). If the "internal" NHL mutations result in decreased protein stability and these NHL patients are homozygous, does this not just kill the level of the protein in the cell. That by extension would of course decrease the binding properties of TREM2, which is the same effect as seen by the AD mutations. It's more pronounced in NHL, but it is fundamentally the same effect. Under this paradigm, both mutations do the identical thing (diminish key TREM2 function) but in different ways. Again, it is the genetic dosage that is driving the severity and decides if the patient will develop NHL or AD. The authors are encouraged to discuss these points.

7) One always worries that the results from such experiments based on individual domains would yield different results from the full protein. Can the authors discuss why they believe that this is or is not the case?

8) The crystal structure appears to be of high quality. However, hydrogen bonds cannot be determined with any certainty at 3.1Å resolution. Please remove all discussion of these structural features.

9) The conclusion "We therefore conclude that GAGs likely orient or cluster TREM2 to mediate signaling upon stimulation by phospholipids. Alternatively, phospholipids may function to orient TREM2 on the cell surface for proper presentation to GAGs and/or additional as of yet undefined protein surface receptors" is not rigorously supported by the data presented.

We urge the authors to consider a thorough rewrite, focusing the manuscript on a coherent, consistent set of results or interpretations, and adding (or redoing) crucial experimental results as outlined above.

[Editors' note: further revisions were requested prior to acceptance, as described below.]

Thank you for resubmitting your work entitled "Neurodegenerative disease mutations in TREM2 reveal a functional surface and distinct loss-of-function mechanisms" for further consideration at *eLife*. Your revised article has been favorably evaluated by Richard Aldrich (Senior editor), a Reviewing editor, and three reviewers.

The manuscript has been improved but one of the reviewers asked a question:

In new Figure 2 the NHD variant s show a large amount of HMW bands consistent with aggregation. Were the samples boiled prior to SDS-PAGE? In many cases boiling of some samples artificially induces aggregation of membrane (though rare with single pass transmembrane proteins). So, if they were boiled it would be nice to see a gel where the sample is not boiled but simply heated to 55 C and then loaded.

---

## [Author Response]

*Summary:*

*This is a biophysical /structural/functional study of the TREM2 ectodomain both from mutant and wild-type TREM2 associated with AD or NH/PLOSL. The bottom line from these studies is the AD associated variants are on the surface of the ectodomain and probably regulate binding of ligands and the PLOSL/NH variants are buried and probably affect folding and maturation as previously discussed. The structural studies appear to be of high quality. However, the other experiments are not optimally or poorly performed, poorly presented, and often over-interpreted, or misinterpreted. The overall recommendation is to re-perform experiments to bring them up to publication quality (e.g., cleanly interpretable gels and analytical SEC traces) or include proper controls, eliminate redundant or irreproducible results, and eliminate unneeded results, as detailed below.*

*Essential revisions:*

*1) General/stylistic comment the reference to the Trem2 Ig domain can be confusing. Often the authors refer to an experiment with this domain and then quickly move on to the full-length protein and one has to pay close attention to figure out which was being used. For the reader it would be useful to call the secreted domains something other than TRME 2 Ig which sounds like an immunoglobulin fusion. Maybe just call them soluble TREM2, trem2 ectodmains, or TREM2 (x-Y) x-y being residues of the fusion proteins.*

We have decided to call that construct TREM2 ectodomain and have changed the text and figures accordingly.

*2) Figure 2. There are several concerns:*

*It would be nice to see the Western blots in the primary data not supplemental. Surface expression should probably be normalized to total protein expression. Immunofluorescence of the cells should show intracellular accumulation of certain mutants that aggregates and this would be a nice addition and simple to do.*

We have made the suggested changes and several other revisions to Figure 2 to make it easier to interpret. First, we have added a schematic on the left side of each portion of the figure to highlight which TREM2 construct is being used in each set of experiments. Second, western blots have been moved into the main figure (Figure 2). Third, we performed a western blot to show total protein expression in the cells used for the flow cytometry analysis in Figure 2 (this is Figure 2). One can easily see that all WT, NHD mutants, and AD risk variant TREM2s are all expressed in similar amounts in these cells; in fact, we detect higher expression of the NHD mutants (Y38C, T66M, V126G). Fourth, we carried out immunofluorescence and confocal imaging experiments on non-permeabilized and permeabilized cells for WT TREM2, a representative NHD mutant (Y38C), and AD risk variant (R47H). In non-permeabilized cells, we found that the surface staining pattern was similar for WT, R47H, and Y38C TREM2 (Figure 2). However, we found that the Y38C TREM2 NHD mutant displayed diffuse staining in permeabilized cells (compared to R47H and WT) suggesting that there is a lot of this mutant that is misfolded and does not get to the surface.

*The statement "WT hTREM2 Ig domain consistently eluted as a monomer with >95% purity" (based on Figure 2) is not supported by the data. The wt SEC trace shows considerable aggregation, a significant void-volume peak, and shouldering. The PAGE analysis in Figure 2 suggests that this SEC trace is the preparative trace, with contaminating proteins forming aggregates. Please rerun the SEC traces after final purification, and include a complete set of comparable SEC analyses for all the mutants discussed. Also, comparative reduced/non-reduced PAGE is more informative about folding for this intra-chain disulfide-linked protein. Please provide analytical SEC traces for all of the mutants discussed; this is more informative than the PAGE results shown. Why are Westerns shown in Figure 2, and not PAGE gel results? The PAGE results do not necessarily show immature glycosylation – this usually needs comparative EndoH sensitivity analyses to draw that conclusion rigorously.*

As requested, we re-ran all of the SEC analysis for WT and TREM2 AD risk variant ectodomains (R47H, R62H, N68K, D87N, T96K) as requested. We first purified these proteins by preparative SEC, and then ran the analytical SEC as suggested. The resulting chromatograms display a single monomeric peak, as predicted. We show the example of WT TREM2 ectodomain in Figure 2; identical chromatograms for the AD risk variants were obtained and are shown in Figure 2—figure supplement 1. We also show the SDS-PAGE analysis of these TREM2 ectodomains under reduced and non-reduced conditions as requested and confirm that no improper intermolecular disulfides are formed (Figure 2). Additionally, we include the preparative SEC chromatograms for WT, AD risk variant, and NHD mutant TREM2 ectodomains in Figure 2—figure supplement 1. The TREM2 NHD mutant ectodomains are not soluble enough to obtain enough material to run the peaks from the preparative run on an analytical run. Thus, for the NHD mutants we provide the preparative chromatograms (Figure 2—figure supplement 1) along with western blot analysis, and show that they elute as oligomers. We also show western blot analysis (non-reduced vs reduced) of the TREM2 NHD mutant ectodomains (Y38C, T66M, V126G) media supernatants and show that these variants all run has higher MW oligomers under non-reduced conditions, suggesting that they are misfolded and contain aberrant intermolecular disulfide bonds (Figure 2).

*The results in Figure 2 are inconsistent. Anti-FLAG analyses show consistent levels of cell-surface expression. The authors are making the assumption that the antibodies used are conformational, so the strictly-rigorous conclusion is that these mutant proteins adopt a different conformation, maybe unfolded to some degree, but one that is poorly recognized by these antibodies – cell-surface expression levels appear completely comparable by anti-FLAG. This conclusion also conflicts with the Figure 3 results, which show a "significant" change in the structure of the R47H mutant by CD, which apparently is undetectable by the antibodies used in Figure 2.*

Figure 2 show flow cytometry analysis of FLAG-TREM2 surface expression probed two different ways; Figure 2 uses an anti-FLAG antibody, which is non-conformational and will recognize both correctly folded and misfolded FLAG-TREM2, whereas Figure 2 uses a conformational anti-TREM2 antibody which likely only recognizes properly folded TREM2. We find that all of the TREM2 NHD mutants are cell surface expressed (although expression is decreased for V126G), however these mutants are not strongly recognized by the anti-TREM2 antibody. This is consistent with the observations in Figure 2, and we clarify this by including the schematics in Figure 2. Also, we detect a change in the CD spectrum which is statistically “significant”; however, this does not likely translate into a change in structure large enough to disrupt antibody binding). By significant, we meant “statistically significant”. We have now inserted that word. We now state in the text “Thus the R47H mutation in TREM2 seems to induce a small, but measurable, conformational change while the R62H and T96K mutations do not induce any structural changes detectable by CD.”

*3) Figure 3. There are several concerns:*

*Unless the mutants are confirmed to be monodisperse (free of aggregates, confirmed by analytical SEC analyses, which needs to be shown), then all of the biochemical analyses reported in Figure 3 are not useful. The CD analyses show nice cooperative transitions, so I expect many of these preparations are monodisperse, but this needs to be confirmed. The redundancy in the current presentation by using multiple approaches to assess stability does not improve impact, and are meaningless if the tested protein is aggregating. Overall, there might be subtle differences here but in the current form they do not add a whole lot to the paper.*

Figure 3 can be deleted, and the information in Figure 3 can be combined into frames E and F.

As we have demonstrated with the chromatograms and SDS-PAGE analysis in Figure 2, that these samples are monodisperse. We have consolidated Figure 3 as suggested and have moved the other parts to Figure 3—figure supplement 1.

*Why is the gel shown in Figure 1—figure supplement 2. (a) an immunoblot? Wouldn't a straight PAGE gel be cleaner? The proteins should be available to perform this analysis.*

We are confused by this concern. The point of this supplementary figure was to show that EndoHf treatment (which removes all glycans except the terminal one attached to the Asn) of TREM2 ectodomains for crystallization results in uniform digestion of glycans (so that there is a single band for TREM2) similar to treatment with PNGaseF (which removes all glycans). We have previously shown the SDS-PAGE for these samples in our technical and crystallization papers (Kober et al., Prot Expr Purif 2014, Figure 2; and Kober et al., J Vis Exp 2015, Figure 1).

*4) Figure 4. There is really no control here. If one takes a protein that is known to bind only to a receptor and does the same study what happens? So, while internal comparisons between the trem2 variants is valid. I think some sort of control for non-specific interactions is needed. Maybe just an Fc domain with that tag or just some other protein ectodomain. Overall, are any of these differences significant? There are no positive and negative binding controls shown, and the results could easily simply represent non-specific binding to sticky lipidic species.*

We have restructured Figure 4 and added schematics to more explicitly describe the experiments being presented. The control for these experiments is now shown in Figure 4 and Figure 4. We have used the TREM-like proteins TREML2 and TREML4 as controls to show that the binding to cells that we observed is specific. TREML2 and TREML4 are both in the TREM family, and are predicted to have the same fold, and in addition, their ectodomains are highly basic (as denoted by the high calculated π values [with the exception of mTREML4]). However, by structure-based sequence analysis, these proteins would not contain the basic patch residues conserved only in TREM2. As shown in Figure 4, these ectodomains (TREML2 and TREML4) do not bind to N2A cells, whereas TREM2 robustly binds these cells.

*Also, what is the point of the rest of this figure, comparing to TIM4? It should be obvious that the PS binding site on TIM4 bears ZERO structural resemblance to the feature being highlighted on TREM2.*

There are three Ig domains that directly bind lipids that have been structurally characterized. TIM4, TIM1, and CD300a all use the same motif and binding site to accomplish this. The point of the figure is that TREM2 does not contain these structural determinants and therefore likely engages lipids directly via a unique manner.

*5) It is very difficult to make sense of the cell-based binding studies, in that many of the results appear wildly inconsistent. For instance, compare the binding of wt mTREM2/SA-PE to unmodified CHO cells in Figure 4 vs. 4I, or the level of hTREM2/SA-PE binding to untreated THP-1 cells in Figure 4 vs. 5G. These analyses appear too irreproducible to be reliable, suggesting that they may be artifactual.*

We have made the presentation of these results more clear with new schematics added to Figure 4 and have also taken care to remove any unneeded results. It is true that the magnitude of binding of any reagent to cell surface receptors will vary depending on any variable that can effect the concentration of target receptors on the cell surface (e.g., cell passage number, etc.). These effects are even more pronounced when a lower-affinity reagent (as compared to an antibody) is used. In all of our cell-based binding studies, we have taken care to run all of the comparative experiments with the same batches (passages) of cells. In addition, all of these experiments were run at least three times each. The general binding trends were always the same. Finally, we did the experiments comparing the TREM2 AD risk variants (R47H, R62H, and T96K) in three different orientations, i.e. as TREM2/SA-PE tetramers directly binding cells, as TREM2 monomers directly binding cells, and finally monomeric WT TREM2 and monomeric TREM2 risk variants competing for binding with WT TREM2/SA-PE tetramers). We obtained the same trend regardless of the orientation of the experiments. So, we feel that these experiments have been carried out quite exhaustively and are entirely reproducible.

*6) From a genetics perspective, NHL cases are homozygous and AD cases are heterozygous. Indeed, some of the same mutations that cause NHL in their homozygous form lead to AD in the heterozygous form. While the authors do provide convincing evidence that the NHL and AD mutations are located in different parts of the 3D structure of the protein, they seem to ignore the obvious dosage effect. (It is discussed briefly in the last paragraph, but this is far too important an effect to be dismissed lightly). If the "internal" NHL mutations result in decreased protein stability and these NHL patients are homozygous, does this not just kill the level of the protein in the cell. That by extension would of course decrease the binding properties of TREM2, which is the same effect as seen by the AD mutations. It's more pronounced in NHL, but it is fundamentally the same effect. Under this paradigm, both mutations do the identical thing (diminish key TREM2 function) but in different ways. Again, it is the genetic dosage that is driving the severity and decides if the patient will develop NHL or AD. The authors are encouraged to discuss these points.*

This is an excellent point and not one to be overlooked, as the reviewers have stated. We have added more on this issue in the last paragraph of the Discussion, and added this to the summary figure in Figure 6.

*7) One always worries that the results from such experiments based on individual domains would yield different results from the full protein. Can the authors discuss why they believe that this is or is not the case?*

It is very common for the binding properties of single pass transmembrane receptor ectodomains to be analyzed free from their transmembrane regions. Examples of this would be Chris Garcia’s structural and biophysical work on cytokines and cytokine receptors (e.g., IL-2 and Il-15, Ring et al., Nat Immunol 2012; IL-13 and IL-4, LaPorte, Cell 2007), interferons and their receptors (e.g., IFNa, Thomas et al., Cell, 2011). In all of these cases, the binding of ligand to isolated receptor ectodomains validated cellular experiments utilizing the full protein. In our case, TREM2 consists of a single Ig domain which is connected to the transmembrane domain by a long (40 a.a.) linker which is likely unstructured. Furthermore, TREM2 is presented on the cell surface as a 1:2 complex with DAP12 dimer (see Figure 1). This has been established by Kai Wucherpfenning’s group (Feng et al., PLOS Biol 2006; Call et al., Nat Immunol 2010). Thus, the current literature would suggest that TREM2 is expressed on the cell surface as in a 1:2 ratio with DAP12, with the single Ig domain tethered to the surface by a long, unstructured linker. Studies of the isolated Ig domains are extremely valid in this case.

*8) The crystal structure appears to be of high quality. However, hydrogen bonds cannot be determined with any certainty at 3.1Å resolution. Please remove all discussion of these structural features.*

Discussion of hydrogen bonds has been removed as requested.

*9) The conclusion "We therefore conclude that GAGs likely orient or cluster TREM2 to mediate signaling upon stimulation by phospholipids. Alternatively, phospholipids may function to orient TREM2 on the cell surface for proper presentation to GAGs and/or additional as of yet undefined protein surface receptors" is not rigorously supported by the data presented.*

We have more explicitly and cautiously explained this statement. We have replaced the above passage with the following: “Since the R47H risk variant is able to directly bind phospholipids, but does not signal in cellular assays of TREM2 signaling triggered by plated phospholipids (Wang et al., 2015b), we suggest that TREM2 interaction with GAGs in cis is likely required to orient or cluster TREM2 to mediate signaling upon stimulation by phospholipids.” These observations are consistent with our current data and others published data.

[Editors' note: further revisions were requested prior to acceptance, as described below.]

*The manuscript has been improved but one of the reviewers asked a question:*

*In new Figure 2 the NHD variant s show a large amount of HMW bands consistent with aggregation. Were the samples boiled prior to SDS-PAGE? In many cases boiling of some samples artificially induces aggregation of membrane (though rare with single pass transmembrane proteins). So, if they were boiled it would be nice to see a gel where the sample is not boiled but simply heated to 55 C and then loaded.*

From our observations, boiling does not induce aggregation in these proteins. For all of the samples involving solubilizing of full-length TREM2 variants, cells were harvested by washing in cold PBS+EDTA, spun down, and then solubilized in detergent (PBS + 0.1% Triton X-100). Next, the insoluble particulate (membrane aggregates, organelles, etc.) were pelleted by centrifugation, and the solubilized supernatant was run on the gel. For the samples in Figure 2, samples were boiled less than 5 minutes, and here we found that we predominantly observed HMW oligomers only for the NHD variants (and a little in the T96K AD risk variant. However, if we boil the samples longer (>10 min), then all samples collapse to single band corresponding to monomeric TREM2. These are the samples we show in Figure 2. Since Figure 2 is cropped, we include below a scan of the full western blot, so you can see that there is only a single band observed for each of the TREM2 WT, NHD mutants, and AD risk variants. In addition, if boiling induced aggregation, we would expect to observe it nearly equally for WT and all variants. Taken altogether, we do not believe that boiling induces aggregation in the TREM2 proteins we analyze here. Instead, we believe that the aggregation we observe in these western blots shows in Figure 2 are indicative of misfolding and covalent oligomers (formed by aberrant intermolecular disulfides) for the NHD mutants, which is consistent with our other data. However, if the reviewers would still like to see samples prepared by heating to 55 degrees as described above, then we can perform that experiment if needed.